# Effectively Learning Initiation Sets in Hierarchical Reinforcement Learning

**Akhil Bagaria***
Brown University
Providence, RI, USA.
akhil_bagaria@brown.edu

**Ben Abbatematteo***
Brown University,
Providence, RI, USA.
abba@brown.edu

**Omer Gottesman**[†]
Amazon,
New York, NY, USA.
omergott@gmail.com

**Matt Corsaro**
Brown University,
Providence, RI, USA.
matthew_corsaro@brown.edu

**Sreehari Rammohan**
Brown University,
Providence, RI, USA.
sreehari_rammohan@brown.edu

**George Konidaris**
Brown University,
Providence, RI, USA.
gdk@cs.brown.edu

## Abstract

An agent learning an option in hierarchical reinforcement learning must solve three problems: identify the option's subgoal (termination condition), learn a policy, and learn where that policy will succeed (initiation set). The termination condition is typically identified first, but the option policy and initiation set must be learned simultaneously, which is challenging because the initiation set depends on the option policy, which changes as the agent learns. Consequently, data obtained from option execution becomes invalid over time, leading to an inaccurate initiation set that subsequently harms downstream task performance. We highlight three issues—data non-stationarity, temporal credit assignment, and pessimism—specific to learning initiation sets, and propose to address them using tools from off-policy value estimation and classification. We show that our method learns higher-quality initiation sets faster than existing methods (in MINIGRID and MONTEZUMA'S REVENGE), can automatically discover promising grasps for robot manipulation (in ROBOSUITE), and improves the performance of a state-of-the-art option discovery method in a challenging maze navigation task in MuJoCo.

## 1 Introduction

Temporal abstraction, which is crucial for scaling RL to long-horizon problems [Konidaris, 2019, Sutton et al., 2022], is elegantly captured using the options framework [Sutton et al., 1999]. Unlike primitive actions that only last for a single timestep, options execute a policy until some termination condition is met. General purpose agents must possess many such options to solve a wide variety of tasks [White, 2017], but increasing the size of the action set raises both the complexity of learning and the branching factor of search [Littman et al., 1995]. This necessitates a mechanism for locally pruning irrelevant options: most options do not apply in a given state and the agent can lower its effective branching factor by ignoring them [Precup et al., 1998, Wen et al., 2020]. Fortunately, the options framework includes the concept of *initiation sets* which captures the idea of local applicability: an option can only be executed from states inside the initiation set.

It is natural to view an initiation set as the set of states from which option execution is likely to succeed; in that case, learning it is straightforward when the option's policy is *fixed*: run the

---

*Equal Contribution

[†]Work done while at Brown University.

37th Conference on Neural Information Processing Systems (NeurIPS 2023).

option from different states, record the outcome, and learn a classifier that predicts whether option execution will succeed at a given state [Konidaris and Barto, 2009, Kaelbling and Lozano-Pérez, 2017]. However, this is complicated by non-stationarity during learning: as the option policy changes, previously recorded outcome labels become invalid. How should the initiation classifier change in response? Another challenge is that initiation set learning suffers from a *pessimistic bias*: the option policy mostly improves for states *inside* the initiation set because option execution only occurs from such states. Therefore, if a state is excluded from the initiation set early during learning, it will likely remain excluded even as the policy improves over time. This pessimistic bias causes initiation sets to mostly shrink and rarely expand [Bagaria et al., 2021a], even in the face of policy improvement; this is potentially catastrophic for learning since the agent prematurely gives up on options in favor of primitive actions [Harb et al., 2018].

To address the issue of non-stationarity, we introduce the Initiation Value Function (IVF), a general value function (GVF) [Sutton et al., 2011, White, 2015] that predicts the probability that option execution will succeed from a given state. Since the IVF is purely predictive, it can be learned using tools from off-policy policy evaluation (OPE) [Voloshin et al., 2021] and does not interfere with the option's main task of maximizing its internal reward function [Sutton et al., 2023]. Unlike the classification approach described above, the IVF adapts to a changing policy: as the policy improves over time, so does the IVF's estimates of its probability of success. We show how the IVF can be used as a direct estimate of the option's initiation set or as input to a *weighted* classifier that accounts for changes in the option policy.

To address the pessimistic bias of learning initiation sets, we expand the criterion for including a state in the option's initiation set: in addition to states from which option execution is likely to succeed, we also include states for which the option policy is most likely to *improve*. By identifying and adding these states to the initiation set, we mitigate the pessimistic bias and prevent initiation sets from collapsing as the option is learned.

For evaluation, we first measure how accurately we can deduce an option's initiation set in MINIGRID-FOURROOMS and the first screen of MONTEZUMA'S REVENGE. Then, we demonstrate that our proposed methods can effectively identify promising grasps in challenging robot manipulation problems in ROBOSUITE. Finally, by integrating our method for learning initiation sets into an existing option discovery algorithm, we solve a maze navigation problem in MuJoCo that baseline agents are unable to.

## 2   Background and Related Work

As is standard in HRL [Barto and Mahadevan, 2003], we consider problems that can be modeled as Semi Markov Decision Processes (SMDPs) $\mathcal{M} = (\mathcal{S}, \mathcal{A}, \mathcal{R}, \mathcal{T}, \gamma)$, where $\mathcal{S}$ is the state space, $\mathcal{A}$ is the action space (which contains options and primitive actions), $\mathcal{R}$ is the reward function, $\mathcal{T}$ is the transition function and $\gamma$ is the discount factor. An option $o \in \mathcal{O} \subseteq \mathcal{A}$ models temporally extended behaviors; $o = (\mathcal{I}_o, \pi_o, \beta_o)$, where $\mathcal{I}_o$, the initiation set of the option, describes the states from which the option can be executed, $\beta_o$ is the termination or subgoal region and $\pi_o$ is the policy [Sutton et al., 1999]. When using RL to learn the option policy, an internal option subgoal reward function $\mathcal{R}_o$ and timescale $\gamma_o$ completes the subtask description of the option and is used to train $\pi_o$ [Precup, 2000, Barto and Mahadevan, 2003, White, 2017].

An important problem in HRL is that of **option discovery**, where algorithms focus on identifying option termination conditions (subgoals) and usually assume that all options apply everywhere [Dayan and Hinton, 1993, Bacon et al., 2017, Eysenbach et al., 2019, Machado et al., 2017]; as previously discussed, this is not a scalable strategy for developing general-purpose agents. We hypothesize that well-learned initiation sets will improve the quality of options, regardless of how they are discovered, and will eventually lead to more performant hierarchical RL agents.

**General Value Functions.**   Value-based RL typically predicts and maximizes a single scalar reward, but a value function can be generalized to predict (and sometimes control [Jaderberg et al., 2017]) the discounted sum of any real-valued "cumulant" that can be computed from the agent's observations [Sutton et al., 2011, White, 2015]. Such Generalized Value Functions (GVFs) efficiently solve the temporal prediction tasks and can represent rich knowledge about the world [Schaul and Ring, 2013].

**Initiation Sets in Robotics and Planning.** Initiation sets are crucial in classical planning [Fikes et al., 1972] as well as task and motion planning (where they are referred to as *preconditions*) [Garrett et al., 2021], but they are usually designed by a domain expert [Kaelbling and Lozano-Pérez, 2017]. In robotics (*affordances*), they are usually programmed to avoid collisions [Xu et al., 2021] or to correspond to relevant object attributes [Şahin et al., 2007, Huang et al., 2023]. In control theory (*regions of attraction*), they denote states where policies are provably stable [Tedrake, 2009, Ames and Konidaris, 2019]. While our proposed methods do not come with finite-time convergence guarantees, they are arguably more general and scalable because they do not assume pre-programmed policies or additional structure such as objects and robot dynamics.

Techniques for learning initiation sets fall into 3 categories: classification, value-based and end-to-end.

**Classification approach.** When initiation sets are learned, it is most often framed as binary classification [Konidaris and Barto, 2009, Bagaria and Konidaris, 2020, Khetarpal et al., 2020, Bagaria et al., 2021b]. This approach is sound when the option policy is fixed [Konidaris et al., 2018] or when the affordances correspond to primitive actions [Khetarpal et al., 2020]. But, this approach does not scale to the continual learning setting [Ring, 1995, Mcgovern, 2002] where new, temporally extended options keep getting discovered and their policies and initiation sets must be learned simultaneously.

**Value function approach.** SayCan grounds large language models using value functions of skills trained using offline RL [Brohan et al., 2023]. Since the policies are pre-trained, they do not confront the challenges of jointly learning policies and their initiation sets discussed in this paper. Nica et al. [2022] learn affordances via Monte Carlo estimation (which is equivalent to binary classification under a restrictive $0/1$ reward function), but do not account for the pessimistic bias or the temporal structure in the initiation set learning problem. Furthermore, both SayCan and Nica et al. [2022] design the option reward function so that the resulting value function can be interpreted as an affordance/initiation probability; this is a strong restriction because RL is famously sensitive to reward design [Randløv and Alstrøm, 1998]. Our method allows the designer to pick the reward function appropriate for their task and the initiation function is learned using the appropriate cumulant using policy evaluation. Relay Networks [Kumar et al., 2018] learn initiation regions by thresholding the critic, but have difficulty picking a threshold because it is not possible to interpret arbitrary value functions as initiation probabilities.

**End-to-end approaches.** IoC [Khetarpal and Precup, 2019] learns initiation sets in the Option-Critic (OC) framework; GrASP [Veeriah et al., 2022] learns affordances that are useful for Monte Carlo Tree Search (MCTS). These are promising gradient-based approaches to learning affordances, but are specific to OC and MCTS respectively. Furthermore, they both learn affordances that maximize *task* reward, we instead focus on learning options that each represent their own subtask [White, 2017] and could later be composed to solve many downstream tasks [Barreto et al., 2019].

## 3 Effectively Learning Initiation Sets

The problem of learning initiation sets can be naturally framed as training a classifier where states on trajectories that achieve the subgoal are labeled as positive examples, and states on trajectories that fail to achieve the subgoal are labeled as negative examples. A probabilistic classifier trained on this data will predict the probability that an option will succeed from a given state, which is exactly the desired semantics of an initiation set. However, when initiation sets and policies have to be learned together, such a classifier is no longer a good estimator, for three reasons:

**Data non-stationarity.** The first reason is data non-stationarity, which refers to the phenomenon that previously collected training examples become invalid during the course of learning, but continue to impact classifier training. Consider a trajectory $\tau_0$ obtained by rolling out an option policy $\pi_o^{t_0}$ at time $t_0$. If the policy is in the early stages of training, $\pi_o^{t_0}$ will likely fail to reach the subgoal region $\beta_o$ and $\tau_0$ will be labeled as a negative example ($Y(s) = 0, \forall s \in \tau_0$). At some future time $t_1$, the policy might improve, but the states along $\tau_0$ will continue to have a negative label, which means that they will continue to be outside the initiation set. As a result, we will fail to capture the growing competence/reachability of $\pi_o^{t>t_0}$. A naïve solution to this problem would be to discard old training

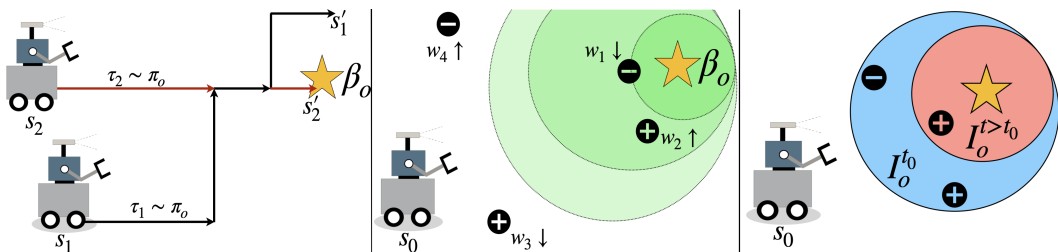

Figure 1: Learning an initiation set for an option targeting subgoal $\beta_o$ (star). *(left)* Temporal structure in initiation set learning: even though $\tau_1$ (black) fails to reach the goal, states along it should get credit because of the portion shared with the successful trajectory $\tau_2$ (red); this does not happen using a simple classification approach. *(middle)* the initiation value function (IVF) $\mathcal{V}_o$ (Sec 3.1) is shown in green: darker the shade, higher the value; classification examples should be re-weighted to account for a changing policy. *(right)* Pessimistic bias causes initiation sets to shrink (blue → red) over time.

examples, but not only would that be sample-inefficient, it is also unclear at what point a training example is "old enough" to be discarded.

**Pessimistic bias.** The second reason, pessimism, refers to the fact that once states are deemed to be outside the initiation set, they are unlikely to switch labels because the option has low probability of being executed from them, and so policy improvement from these states is unlikely (Figure 1 *(right)*).

**Temporal Structure.** The third reason is that classification (or equivalently, Monte Carlo learning [Sutton and Barto, 2018]) does not exploit the temporal structure in the initiation set learning problem. Figure 1 *(left)* shows two trajectories sampled from the same option's policy: $\tau_1$ reaches the goal and $\tau_2$ does not. Any form of Monte Carlo estimation using this data would assign an initiation probability of 0 to most states in trajectory $\tau_1$. However, this does not assign enough credit to most of the good decisions made along $\tau_1$—the only way to achieve that would be via *temporal bootstrapping*. This form of generalization is not possible when using classification.

To address non-stationarity and to exploit temporal structure, we formulate a new general value function, the *Initiation Value Function* (IVF; Section 3.1), learned using temporal difference (TD) methods [Sutton, 1988]. To address pessimism, we augment the initiation set with states from which the option policy is most likely to *improve* (Section 3.3).

### 3.1 Initiation Value Function (IVF)

An option $o$'s initiation set $\mathcal{I}_o$ is the set of states from which the option policy $\pi_o$ can succeed with high probability: $\mathcal{I}_o = \{s : \mathbb{P}(s' \in \beta_o | S = s, A = o) > T\}$ ($T$ is a predefined threshold). Treating the success condition of the option as a cumulant $c_o : s \to \{0, 1\}$ and setting the timescale $\gamma_{c_o} := 1$, the corresponding GVF

$$\mathcal{V}^{\pi_o}(s_t) = \mathbb{E}_{\pi_o}\Big[ \sum_{t=0}^{t=H_o} c_o(s_{t+1})|s_{t+1} \sim \mathcal{T}(s_t, \pi_o(s_t))\Big] = \mathbb{P}(c_o = 1 | S = s_t, O = o)$$

represents the initiation probability at state $s_t$ ($H_o$ is the option horizon). Note that $\mathcal{V}^{\pi_o}$ is different from $V_o$, the value function used by the option policy for control; this is because $\pi_o$ maximizes an arbitrary reward function $\mathcal{R}_o$ and so the resulting value function $V_o$, which approximates the value of the optimal option policy $V^{\pi_o^*}$, cannot be interpreted as an initiation probability.

**Using the IVF as an initiation set.** To directly use the IVF as the initiation set, we need to pick a threshold $T$ above which a state is deemed to be in the option's initiation set. Previous work [Kumar et al., 2018] reused the option value function $V_o$ as the IVF, but had to develop heuristic schemes to pick a threshold. But since $\mathcal{V}^{\pi_o}_\phi$ outputs an interpretable number between 0 and 1 (we use a sigmoid layer at the end of $\phi$ to enforce this), it is easy to threshold, regardless of $\mathcal{R}_o$.

**Learning the IVF.** Computing the probability of success of a given policy $\pi_o^t$ is equivalent to the problem of off-policy policy evaluation. Our approach to policy evaluation *must* be sample-efficient—if it is not, the option policy will change before we are able to sufficiently evaluate it [Sutton et al.,

2007]. Furthermore, the initiation cumulant $c_o$ is a sparse binary function, which makes learning with the Monte Carlo estimator high variance and sample inefficient. This once again underscores the importance of exploiting temporal structure of the problem using TD-learning (we use TD(0)) to estimate $\mathcal{V}_\phi^{\pi_o}$, which unlike a classifier, is able to propagate value from partial trajectories (as illustrated in Figure 1 *(left)*). In this view, using a classifier is equivalent to using a Monte Carlo estimate of the IVF: high-variance, sample inefficient, and unable to bootstrap.

## 3.2 Combining Classification and the Initiation Value Function (IVF)

In Section 3.1 we connected the initiation set to a GVF, which allows us to use all the tools of value-based RL [Sutton and Barto, 2018] to learn the initiation set. However, the classification approach mentioned earlier has some attractive qualities: specifically, classification is easier than regression [Bishop and Nasrabadi, 2006] and the supervised learning community has developed powerful tools for learning classifiers using neural networks and cross-entropy loss [Goodfellow et al., 2016]. To get the best of both worlds, we propose an additional method that combines value estimation and classification.

Recall that binary classification can be described as the process of finding parameters $\theta$ that minimize the cross-entropy loss $\mathcal{L}(\theta)$ in the training data $\mathcal{D} = \{(s_i, Y_i)\}_{i=0}^{|\mathcal{D}|}$ where $s_i$ are the inputs (states in our case) and $Y_i$ are the classification labels (whether option execution was successful from $s$). To deal with non-stationarity, we instead use the *weighted* binary cross-entropy loss $\mathcal{L}_w(\theta)$ which minimizes loss over a dataset $\mathcal{D} = \{s_i, Y_i, w_t(s_i)\}$ where $w_t(s_i)$, a state-dependent weight, represents the desired contribution of the training example to the classification loss [Natarajan et al., 2013].

How should we set the weights $w_t(s)$ of a training example $(s_i, Y_i)$ in a way that reflects the evolving competence of the option policy? Since the IVF evaluates the success probability of the current option policy $\pi_o^t$, we can heuristically use it as a reflection of our certainty in the training example $(s_i, Y_i)$—positively (negatively) labeled states contribute more to $\mathcal{L}_w$ when the IVF prediction is high (low). Specifically, the weighting function is defined via the following simple equation:

$$w_t(s) = Y_s \mathcal{V}(s) + (1 - Y_s)(1 - \mathcal{V}(s)). \tag{1}$$

As the IVF estimates that the option policy is becoming more competent at state $s$, the classifier's uncertainty about a negative example at $s$ will increase, thereby causing that negative example to contribute less to the classifier's decision boundary. Similarly, if the policy degrades at $s$, then the contribution of a positive label at $s$ to $\mathcal{L}_w$ will go down. By weighing training examples in this way, a binary classifier is able to adapt to data non-stationarity that results from a changing option policy.

At the end of every option execution, we therefore recompute weights using Equation 1 and re-train option initiation classifiers using a weighted cross-entropy loss $\mathcal{L}_w$.

**Policy Improvement Prior.** We can use the fact that policies in RL usually improve over time to guide classifier training. To do this, we set the weight of positive examples $w_t(s)$ as 1 throughout training. The underlying idea is that a state which leads to success once is likely to do so again with an even better policy. The weights of negative examples are allowed to vary with the IVF as in Eq 1.

## 3.3 Overcoming Pessimistic Bias

Unsuccessful option trajectories cause the initiation set to shrink. When a state is outside the initiation set, the option can no longer be executed from there. So even if the option could succeed from that state in the future, it remains outside the initiation set and the option ends up with a smaller region of competence than it needs to. This issue, which we call the pessimistic bias of learning initiation sets, can prevent the learning of useful options and lowers the effectiveness of hierarchical RL agents[3].

To mitigate this pessimistic bias, we expand the initiation set to also include states from which policy improvement is most likely:

$$\mathcal{I}_o = \{s : \mathcal{V}_o(s) + \mathcal{B}(s) > T, s \in \mathcal{S}\}.$$

Effectively identifying such states in high-dimensional spaces is the topic of bonus-based exploration in deep RL [Taïga et al., 2019]. We propose two simple approaches:

---

[3]This issue is analogous to non-stationary dynamics in flat MDPs; see Section B in the Appendix.

1. **Competence progress** attempts to capture regions where a policy is either improving or regressing [Şimşek and Barto, 2006, Stout and Barto, 2010, Baranes and Oudeyer, 2013]. This can be computed as changes in the IVF over time: $\mathcal{B}_1(s) = \left| \mathcal{V}_o^t(s) - \mathcal{V}_o^{t-K}(s) \right|$, where $\mathcal{V}_o^t$ is the current IVF and $\mathcal{V}_o^{t-K}$ is the IVF estimate $K$ timesteps ago (obtained using the target network).

2. **Count-based bonus** approach keeps track of the number of times $N(s,o)$ option $o$ has been executed from state $s$. This is then converted into an uncertainty measure: $\mathcal{B}_2(s) = c/\sqrt{N(s,o)}$, where $c$ is a scalar hyperparameter [Strehl and Littman, 2008].

We use count-based bonuses in problems where tabular counts are readily available; otherwise, we use competence progress.

## 4  Experiments

We aim to evaluate whether our methods can improve hierarchical RL. First, we evaluate whether they result in better, more efficiently learned initiation sets. Second, we test if better initiation set learning improves option learning as a whole. Finally, we incorporate our changes into a state-of-the-art skill discovery method and check if resulting agent is able to outperform the baseline in a sparse-reward continuous control problem.

**Implementation Details.**  Option policies are learned using Rainbow [Hessel et al., 2018] when the action-space is discrete and TD3 [Fujimoto et al., 2018] when it is continuous. Following Bagaria et al. [2021a], all options share the same UVFA [Schaul et al., 2015] but condition it using their own subgoals. The IVF is learned using Fitted Q-Evaluation [Le et al., 2019], prioritized experience replay [Schaul et al., 2016] and target networks [Mnih et al., 2015]. The IVF Q-function and initiation classifier are parameterized using neural networks that have the same architecture as the Rainbow/TD3. Each option has a "gestation period" of 5 [Konidaris and Barto, 2009], which means that before the option sees 5 successful trajectories, its initiation set is optimistically initialized to be true everywhere. Since the training data for classification can be severely imbalanced, we use the standard technique of upweighting the minority class [Japkowicz and Stephen, 2002].

### 4.1  Measuring the Quality of Initiation Sets

Before comparing different methods based on task performance, we specifically test the quality of initiation sets learned in MINIGRID-FOURROOMS [Chevalier-Boisvert et al., 2018] and the first screen of MONTEZUMA'S REVENGE [Bellemare et al., 2013]. In both domains, observations are $84 \times 84$ images and the action-space is discrete. For this experiment, we design start states $\mathcal{S}_0$ in each domain—in MINIGRID-FOURROOMS, $\mathcal{S}_0$ is the set of all states (since this is a small tabular domain) and in MONTEZUMA'S REVENGE, $\mathcal{S}_0$ is a series of 100 states scattered across the first screen (more details in the appendix). For the purpose of evaluating our initiation set learning algorithms, at every episode, we sweep through states $s \in \mathcal{S}_0$ and reset the simulator to $s$. Option termination conditions are also hand-defined: in FourRooms, we create options that target the center of each room; in Montezuma's Revenge, we define five options: those that navigate the player to each of the two doors, one that navigates the player to the bottom-right and one to the bottom-left of the first screen and finally an option that attempts to get the key.

**Initiation Set Accuracy.**  At each state $s \in \mathcal{S}_0$, we record the initiation decision made by the learning algorithm as $\hat{\mathcal{I}}_o(s; \theta) \in \{0, 1\}$. We then execute the option policy $\pi_o$ from that state and record whether or not the agent reached the option's subgoal as $Y_s \in \{0, 1\}$. Accuracy at state $s$, for option $o$ is then given by $\mathbb{1}(\hat{\mathcal{I}}_o(s; \theta) = Y_s)$. This process is repeated several times for all options $o \in \mathcal{O}$ and start states $s \in \mathcal{S}_0$. To be faithful to how options are learned in the online RL setting, we only use the trajectory obtained by running the option for updating the initiation classifier and the option policy if the learned classifier returned true at the start state, i.e, $\hat{\mathcal{I}}_o(s; \theta) = 1$ (in which case $\pi_o$ is updated using Rainbow [Hessel et al., 2018]; pseudocode in Appendix C.1).

**Initiation Set Size.**  Although accuracy is a natural evaluation metric, it does not fully capture the quality of the learned initiation set. For example, if the predicted initiation function returns false

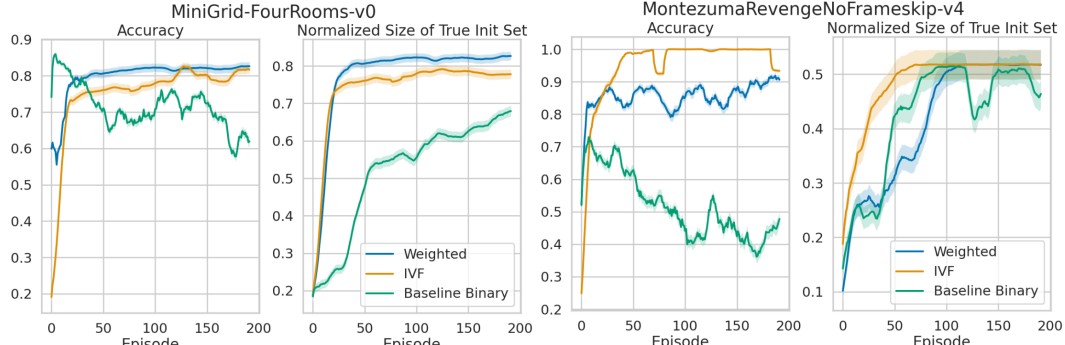

Figure 2: Measuring the quality of initiation sets in MiniGrid-FourRooms (left) and the first screen of Montezuma's Revenge (right). Solid lines denote mean accuracy and initiation set size, shaded regions denote standard error. All curves are averaged over all state-option pairs and 5 random seeds.

everywhere (i.e, if $\mathcal{I}_o(s; \theta) = 0, \forall s \in \mathcal{S}_0$), then the option policy would not get new data to improve. If the policy never improves, its true initiation set could also collapse; while such an initiation learner would register high accuracy, the option would not be useful. As a result, we additionally measure the normalized size of the "true" initiation set $|Y_s|$: this is the fraction of start states $\mathcal{S}_0$ from which Monte Carlo rollouts of the policy succeeds. A well-learned initiation set is not only accurate, it is also as large as possible,[4] reflecting a large region of option policy competence.

**Competing methods.** We compare the accuracy and size of the initiation sets learned by:

- *Baseline binary.* Binary classifier used to learn the initiation set. This is what is used in essentially all prior work and is the only baseline method in this experiment.
- *IVF.* Threshold the Initiation Value Function as discussed in Section 3.1.
- *Weighted.* This is a weighted binary classifier discussed in Section 3.2.

Both the *IVF* and *Weighted* approaches use the competence progress exploration bonus described in Section 3.3, its impact is ablated in Appendix E.

**Discussion of results.** Figure 2 shows that our proposed methods significantly outperform the baseline binary classification approach, which learns classifiers that are less accurate and smaller than those learned using other techniques. Furthermore, the accuracy of the baseline binary classifier *falls* over time; this is because even though the policy improves (and hence the size of the true initiation set increases), the learned initiation set remains small and unable to adapt to an improving policy. By modifying the binary classifier using the weighting scheme described in Section 3.2, we are able to learn more accurate initiation sets that reflect highly competent option policies. Interestingly, the optimistic version of the IVF is necessary to learn good initiation sets in Montezuma's Revenge, but it underperforms compared to the plain IVF in FourRooms; we hypothesize that this is because FourRooms is a small and simple enough domain that an exploration bonus is not strictly necessary.

### 4.2 Robot Manipulation: Identifying Promising Grasps

In the previous section, we showed that our methods can lead to higher quality initiation sets; now we evaluate whether that can in turn improve option learning as a whole. We adopt robot manipulation as a testbed because it is a natural problem setting for evaluating initiation set learning algorithms—for example, consider a robot manipulating a hammer: not only should it choose a feasible, stable grasp on the hammer, but it should also choose one that would allow it to subsequently drive a nail. This problem, referred to as *task-specific grasping* in robotics, and has been studied extensively for a fixed policy [Kokic et al., 2020, Fang et al., 2020, Zhao et al., 2021, Wen et al., 2022, Schiavi et al., 2022]; here we use RL to deduce which grasps are most likely to afford success while simultaneously learning the skill policies. Furthermore, dense reward functions are often necessary to train RL policies in

---

[4]This is similar to the considerations of precision/recall in supervised learning.

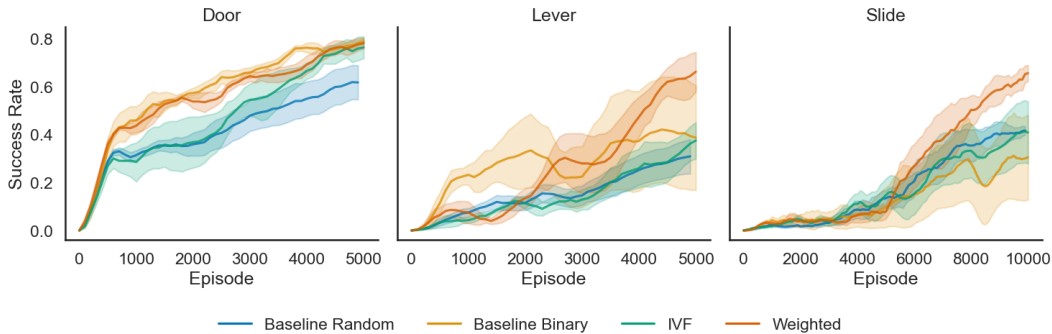

Figure 3: Task success rate for manipulation domains aggregated over 4 random seeds.

robot manipulation domains [Gu et al., 2017]; this section highlights that initiation probabilities can be learned even when the reward function used to train $\pi_o$ is not sparse like the initiation cumulant.

We use three constrained manipulation tasks in ROBOSUITE [Zhu et al., 2020]: opening a door, flipping a lever, and manipulating a sliding mechanism. Each task is modeled as a single option and the policy and initiation set are learned simultaneously. Each environment has 250 possible start states $\mathcal{S}_0$. At the start of each episode, a configuration is sampled from the initiation set learner:

$$ s_0 \sim \frac{\mathcal{I}_o(s)}{\sum_{s' \in \mathcal{S}_0} \mathcal{I}_o(s')}, \forall s \in \mathcal{S}_0; $$

the manipulator is then reset to $s_0$. The continuous observation space is 52-dimensional, the continuous action space is 13-dimensional; we use TD3 for policy learning [Fujimoto et al., 2018].

Our algorithms are compared against two baselines: *Random*, which selects an arm configuration at random at the start of each episode[5], and *Binary* which treats initiation learning as vanilla binary classification. Our algorithms, *IVF* and *Weighted*, employ the optimistic bias discussed in Section 3.3 by adding a count-based bonus to the predicted initiation probability.

Task success rates are shown in Figure 3. The DOOR task is the simplest: most of the grasp candidates afford task success. As a result, the *Random* and *Binary* baselines are competitive with our algorithms. The LEVER and SLIDE tasks are more challenging as the number of promising grasps is significantly lower. In these domains, only the *Weighted* variant is able to consistently recover a policy with success rate above 50%.

Figure 4 shows that our methods can identify good grasp poses without task-specific engineering or human demonstrations. This is impressive given that human data [Mandikal and Grauman, 2021, 2022] or heuristics like image segmentation [Kokic et al., 2017, Rosen et al., 2022] are typically required to learn these affordances efficiently, even in the case of a fixed policy. Additional visualizations and ablations can be found in Appendix D and E.

### 4.3 Improving Option Discovery

Finally, to evaluate whether better option learning can improve option discovery, we integrate our approach into an existing state-of-the-art algorithm: deep skill chaining (DSC) [Konidaris and Barto, 2009, Bagaria and Konidaris, 2020]. DSC learns a collection of options so that the subgoal region $\beta_{o_i}$ of an option $o_i$ is the initiation region $\mathcal{I}_{o_j}$ of another option $o_j$; by learning options that funnel into each other's initiation sets, DSC learns how to sequentially compose options to reliably achieve a goal. It does so by first learning an option that reaches the task goal, then another option that targets the first option's initiation set and so on, until the start-state is inside the initiation set of some option; for more details, please refer to Appendix C.3 and the original papers [Konidaris and Barto, 2009, Bagaria and Konidaris, 2020]. We chose to integrate our techniques with DSC because of its focus on learning initiation sets jointly with options in an online and incremental reinforcement learning setting and because the quality of initiation sets is crucial to its performance and stability [Bagaria et al., 2021a].

---

[5]This baseline is equivalent to an initiation set that is uniformly true everywhere.

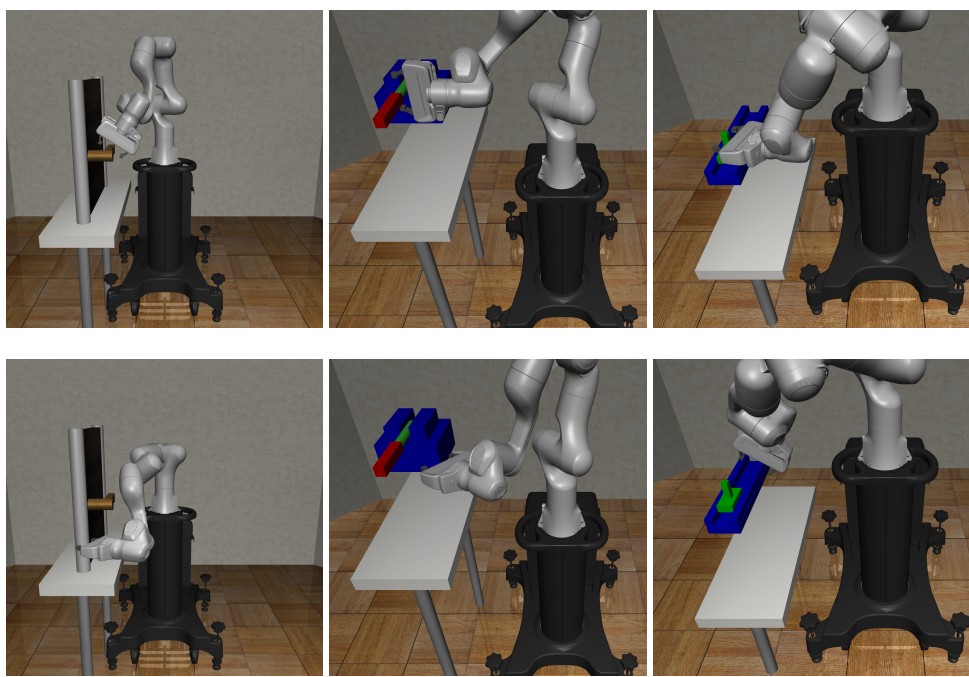

Figure 4: Examples of promising grasp poses *(top row)* in DOOR *(left)*, LEVER *(middle)* and SLIDE *(right)*: these are high probability samples from our initiation function; contrasted with bad grasp poses *(bottom row)*, which are low probability samples and are ruled out by the initiation function.

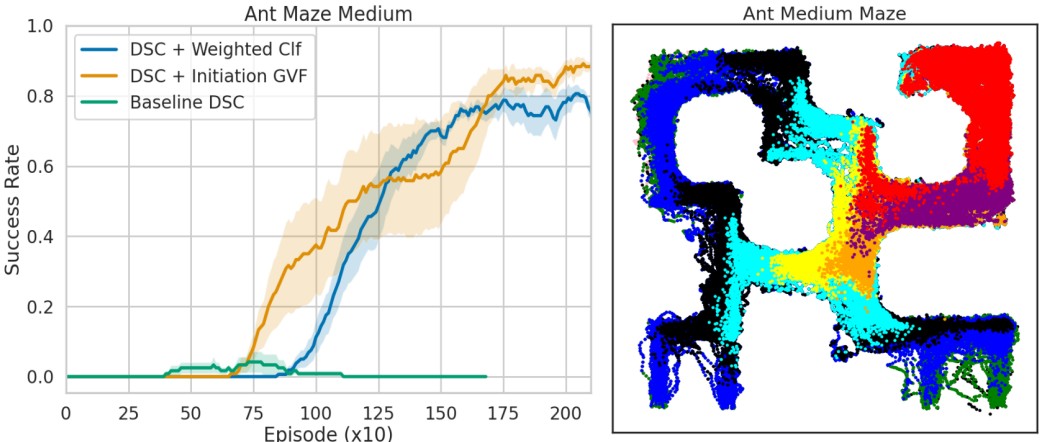

Figure 5: *(left)* Comparing the performance of baseline DSC [Bagaria et al., 2021a] to versions of DSC with our initiation learning techniques. Solid lines denote average success rate over 5 random seeds, shaded regions denote standard error. *(right)* A visualization of initiation sets learned in ANT MEDIUM-MAZE, where the ant robot has to navigate from the bottom-right to the top-left. Each color denotes the initiation set of a different option; although the plot only shows the location of the ant in the maze, the initiation set is learned using the full 30-dimensional state.

**Experimental setup.** We compare baseline DSC to versions that modify the way in which option initiation sets are learned. We use the ANT MEDIUM MAZE environment where the agent gets a sparse terminating reward of $0$ for reaching the goal and $-1$ every other step; each episode lasts a maximum of $1000$ steps [Fu et al., 2020, Todorov et al., 2012]. This is a challenging problem that cannot be solved with flat RL algorithms in a reasonable amount of time. Baseline DSC was able to solve this problem by learning initiation sets defined over a subset of the state ($x, y$ location of the ant) [Bagaria et al., 2021a], but here, we do not assume access to such privileged information—all methods learn initiation sets using a dense neural network that maps the full $30$-dimensional continuous state to an initiation probability.

We used the author implementation [Bagaria et al., 2021a] where option policies are parameterized using goal-conditioned value functions [Schaul et al., 2015]. When using the IVF directly as the initiation set, we mimic the structure of main agent and use a goal-conditioned value function to represent the IVF; the final initiation set is defined as

$$\mathcal{I}_o = \left\{ s : \max_{g \in \beta_o} \mathcal{V}_\theta(s, g) > T, \forall s \in \mathcal{S} \right\}.$$

More details about the task and implementation details can be found in Appendix A and C.3.

Figure 5 *(left)* shows that baseline DSC is unable to solve Medium Ant-Maze; but by changing the way options learn their initiation sets, we are able to solve the problem quickly and reliably. The weighted classifier slightly underperforms the pure IVF approach in terms of mean return, but it has lower variance across runs. Figure 5 *(right)* visualizes the initiation sets learned by the weighted classification approach; the figure makes it clear that different learned options specialize in different parts of the maze.

## 5    Conclusion

Learning initiation sets is a critical component of skill discovery, but treating it as binary classification misses key characteristics of the problem. Specifically, it does not address the non-stationarity that results from a continually changing policy, the pessimistic bias of learning initiation sets online, or the temporal structure in the problem. To address these challenges, we proposed the Initiation Value Function (IVF), a general value function tailored specifically to learning initiation sets. We used the IVF directly via thresholding and also via a weighted binary classifier which adapts to changing option policies. Through our experiments, we showed that our proposals lead to higher quality initiation sets, can lead to faster learning of a single option and boost the performance of an existing skill discovery algorithm.

A limitation of the IVF cumulant proposed in Section 3.1 is that only applies for goal-reaching options, which although is quite general, it is not universal. For example, if the option's task is to maximize the velocity of a robot and there is no specific target velocity, then we could not write down a $0/1$ cumulant that faithfully describes that subtask. Designing cumulants that result in initiation probabilities for general option subtasks is an important avenue for future work.

## Acknowledgements

We thank Mark Rowland and Rafael Rodriguez-Sanchez for useful discussions. We also thank our anonymous reviewers for making thoughtful suggestions that improved the quality of our paper. This research was supported in part by NSF 1955361, NSF CAREER 1844960 to Konidaris and ONR N00014-21-1-2200. The U.S. Government is authorized to reproduce and distribute reprints for Governmental purposes notwithstanding any copyright notation thereon. The content is solely the responsibility of the authors and does not necessarily represent the official views of NSF or the ONR.

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

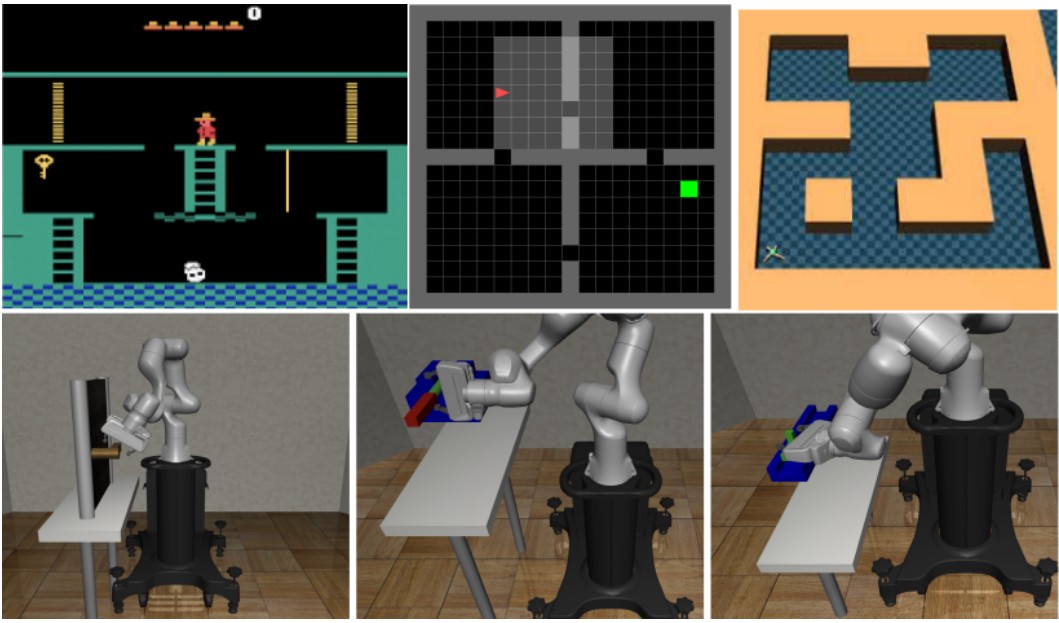

Figure 6: Domains used for our experiments in Section 4. Enumerating left to right from the top-left: first screen of Montezuma's Revenge, MiniGrid FourRooms, Ant Medium Maze, Robosuite Door, Robosuite Lever, and Robosuite Slide.

## A Task Descriptions

**Four Rooms.** This task, which is part of the MINIGRID suite [Chevalier-Boisvert et al., 2018], is an adaptation of the four rooms problem presented in the paper introducing the Options framework [Sutton et al., 1999]. Observations are $84 \times 84$ images of the fully-observable underlying state-space, which contains approximately $19 \times 19$ grid locations and $4$ possible orientations of the player. We defined $4$ options whose termination conditions (subgoals) were to navigate to the center of the $4$ rooms. $\mathcal{S}_0$ was the set of all empty grid locations.

**Montezuma's Revenge.** As is standard in ALE [Bellemare et al., 2013], observations are $84 \times 84$ images, action space is a set of $18$ discrete actions [Machado et al., 2018]. We defined start states scattered across the first room where the player was on the ground (not jumping or falling), was not too close to the skull and did not already have the key in its possession. We also defined $5$ options whose termination conditions are reaching the left door, right door, bottom-right of the first screen, bottom-left of the first screen and getting the key.

**Robosuite manipulation tasks.** Three constrained manipulation tasks were used to study the task-oriented grasping performance of our initiation set learning algorithms: opening a door, flipping a lever, and sliding a knob. The door task was originally implemented in ROBOSUITE [Zhu et al., 2020]; the others were implemented and described in the work of Rosen et al. [2022]. All three are 1-DoF articulated objects which require making and sustaining contact to manipulate. The 52-dimensional observation space consists of the robot's proprioceptive state (joint position, joint velocity, end-effector position, gripper state, tactile data) as well as the object state (object pose, joint position, handle position). The action space employed is operational space control with variable impedance [Martín-Martín et al., 2019]: the agent controls the 6-DoF change in position and orientation of the end-effector, the 6-DoF change in stiffness, and 1-DoF gripper state. Episodes have a maximum length of 250 steps. In each task, $\mathcal{S}_0$ was a set of arm configurations establishing contact with the object; see Section C.2.

**Ant Medium Maze.** The goal location is small region around $(20, 20)$. A state is considered to satisfy a goal if the two have a euclidean distance of 0.5 units or less $R(s, g) = ||s - g||_2 < 0.5$. The agent is evaluated by rolling out the learned policy once every 10 episodes; during evaluation,

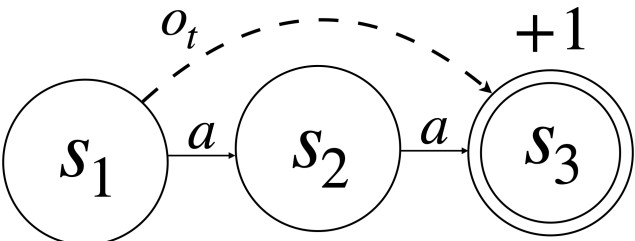

Figure 7: Illustration of the pessimistic bias in a tabular MDP. Action $o_t$ has non-stationary effects, just like an option in HRL. As $o_t$ executed more, the probability of landing in $s_3$ increases, otherwise it leaves the agent in $s_1$.

the agent starts from a small region around $(0, 0)$, during training it starts at a location randomly sampled from the open locations in the maze. The task reward function is $-1$ for non-goal transitions and a terminating reward of $0$ for reaching the goal. Episodes last a maximum of $1000$ steps. The training and evaluation protocol is identical to Bagaria et al. [2021a], except for the fact that we learn initiation sets over the full state.

## B  Pessimistic Bias: Connections to Non-Hierarchical MDPs

MDPs with actions that have non-stationary effects mirror hierarchical MDPs in which the option policies are being learned online. Figure 7 shows such an MDP—there are three states: $s_1, s_2, s_3$, two actions $a$ and $o_t$, and discount factor $\gamma < 1$. Action $a$ takes the agent one state to the right, state $s_3$ yields a terminating reward of $1$, reseting the state to $s_1$. Action $o_t$ has a non-stationary effect: it either leaves the agent at state $s_1$, or takes the agent to $s_3$. The probability with which $o_t$ takes the agent to $s_3$ evolves over time—the more it is executed, the more consistently it takes the agent to $s_3$, similar to an option in HRL whose policy improves with execution.

A Q-learning agent quickly learns $Q(s_1, a) = \gamma$ and $Q(s_1, o_t) < \gamma$ leading to greedy policy $\pi(s_1) = a$. But if the agent executes $o_t$ enough times, it will get higher return $Q(s_1, o_t) = 1 > Q(s_1, a) = \gamma$ leading to the greedy policy $\pi(s_1) = o_t$. Vanilla Q-learning alone would not discover the higher value solution because it does not account for non-stationary effect of $o_t$; some form of exploration is needed for that.

## C  Details about Learning Algorithm

### C.1  Accuracy Experiment

Algorithm 1 is the pseudocode used for the experiments described in Section 4.1. Every episode, every option is executed from every start state in $\mathcal{S}_0$. The result of that execution is recorded as ground-truth $Y_{s,o}(t)$ and stored to later compute the size of the true initiation set $|Y| = \frac{\sum_{s,o,t} Y_{s,o}(t)}{|Y_{s,o}(t)|}$. If the start was predicted to be inside the initiation set by the learning algorithm, then the trajectory generated by rolling out the option policy is used to update the policy and the initiation learner (e.g, IVF, classifier). We report the agreement between the predicted initiations and the ground-truth as an accuracy measurement for that state-option pair. The fraction of start states in $\mathcal{S}_0$ that lead to success is reported as a measurement of the size of the true initiation set.

### C.2  Robot Manipulation

The task-specific grasping problem is typically phrased as identifying grasp poses $g \in \mathbb{SE}(3)$ that afford task success. In practice, the difficulty of this problem is compounded by the fact that, for redundant manipulators, each grasp pose $g$ yields an infinite number of corresponding arm

**Algorithm 1** Accuracy/Size Experiment Procedure

**Inputs**: Option termination conditions $\beta_o, \forall o \in \mathcal{O}$, start states $\mathcal{S}_0$, number of episodes `n_episodes`.
**Outputs**: Accuracy table $A$ and Initiation Size table $S$; both map state-option pairs to a list of booleans.

> Initialize goal-conditioned policy $\pi_\theta : \mathcal{S} \times \mathcal{G} \to a \in \mathcal{A}$.
> Initialize Initiation Value Function (IVF) $\mathcal{V}_\phi : \mathcal{S} \times \mathcal{G} \to \mathbb{R}$.
> Initialize binary classifier for each option $\mathcal{I}_o(s; \psi) \to \{0, 1\}$.
> Initialize replay buffers for Rainbow $B_R$ and IVF $B_I$.
> Initialize buffers to store positive and negative examples for each option's initiation classifier.
> Initialize tables $A$ and $S$ as mapping each state-option pair to an empty list.
> **for** episode $\in$ `range(n_episodes)` **do**
>> **for** start state $s_0 \in \mathcal{S}_0$ **do**
>>> **for** option $o \in \mathcal{O}$ **do**
>>>> Reset simulator to $s_0$.
>>>> Record option $o$'s initiation decision $X = \mathcal{I}_o(s_0; \psi)$.
>>>> Rollout option policy $\pi_o(s_0, g \sim \beta_o)$ to get trajectory $\tau$ and next state $s'$.
>>>> Record whether the option policy reached the goal $Y = s' \in \beta_o$.
>>>> Record accuracy $A[s_0][o]$.`append`($\mathbb{1}(X = Y)$).
>>>> Update ground-truth size table $S[s_0][o]$.`append`($Y$).
>>>> **if** predicted initiation $X = 1$ **then**
>>>>> Add trajectory $\tau$ to policy's replay $B_R$.
>>>>> Relabel trajectory $\tau$ with initiation cumulant $c_0 : \mathcal{S} \to \{0, 1\}$.
>>>>> Add relabeled trajectory to IVF's replay $B_I$.
>>>>> Add trajectory $\tau$ to $o$'s positive/negative example buffer.
>>>> **end if**
>>> **end for**
>> **end for**
>> Sample minibatch and update $\pi_o$ using Rainbow.
>> Sample minibatch and update $\mathcal{V}_\phi$ using TD(0).
>> **for** option $o \in \mathcal{O}$ **do**
>>> Compute weights $w(s)$ for all training examples using Equation 1.
>>> Update $o$'s initiation classifier by minimizing weighted cross-entropy loss.
>> **end for**
> **end for**

configurations (solutions to the inverse kinematics problem). Explicitly, this relation is governed by the manipulator's forward kinematics $f : \mathcal{C} \to \mathbb{SE}(3)$ which maps (typically) 7-DoF configurations of the arm $q \in \mathcal{C}$ to poses in Cartesian space. In practice, only a subset of these configurations for a given grasp pose enable successful manipulation [Schiavi et al., 2022]. As a result, we task the initiation set learning algorithm with choosing start states directly from the space of arm configurations $\mathcal{C}$.

We generate collision-free grasp poses on each object using off-the-shelf grasp generation method GPG [Ten Pas et al., 2017] and corresponding arm poses using IKFLOW[Ames et al., 2022]. We chose to generate 50 grasp poses with 5 random inverse kinematics solutions each yielding a total of 250 starting configurations for each task.

**Reward function.**   The reward functions are implemented as progress toward 1-DoF object joint position goals. The agent receives reward when the current joint position exceeds its previous maximum in a given episode.

**Parameterization.**   As described in Section A, the agent receives proprioceptive and object-state observations and controls the manipulator's end-effector pose and impedance. The learning algorithm employed is TD3 [Fujimoto et al., 2018]. Goal-conditioning is omitted in these experiments as they have a single goal and a single option.

## C.3 Deep Skill Chaining

Deep skill chaining (DSC) [Konidaris and Barto, 2009, Bagaria and Konidaris, 2020] proceeds recursively backward from the goal: the termination region of the first option is the task goal (Line 2, Algorithm 3); the agent first learns an option policy and initiation set for that option. Then, it learns another option whose termination region is the initiation set of the first option; this process continues until there is some option whose initiation set covers the start state. The result is a collection of options that can be sequenced to reliably reach the goal.

Since the initiation sets of options are the subgoals for other options, the entire process is sensitive to the way in which the initiation sets are learned: poorly estimated initiation sets can lead to subgoals that do not improve the agent's ablity to reach the task goal.

Details about line 12 of Algorithm 2 differ based on the method used to learn the initiation set. When using the pure IVF approach, we perform as many minibatch gradient updates as the length of the option rollout; when using weighted classification, we recompute weights using Eq 1 for all training examples and then proceed to minimize weighted cross-entropy loss (3 epochs, batch size 128). When using classification (weighted or unweighted), we boost the contribution of the minority class by the ratio of the size of the majority class to that of the minority class.

---

**Algorithm 2** Robust DSC Rollout

**Inputs.** Skill Chain $\mathcal{O}$
**Hyperparameters.** Option horizon $H$

1: Initialize empty trajectory buffer $\mathcal{B}$
2: **for** each timestep $t$ **do**
3:     Select option $o$ using policy over options $\pi_{\mathcal{O}}(s_t)$
4:     Sample a goal for selected option: $g \sim \beta_o$
5:     Execute option policy $\pi_o(\cdot|g)$ in the environment
6:     Add trajectory $\tau = \bigcup_{i=0}^{H-1} (s_i, o, a_i, s_{i+1}, g)$ to $\mathcal{B}$
7:     **if** final state $s_H$ reached goal $g$ **then**
8:         Add $\tau$ to $o$'s list of positive examples
9:     **else**
10:        Add $\tau$ to $o$'s list of negative examples
11:     **end if**
12:     Refit option $o$'s initiation classifier
13:     Add $\tau$ to replay buffer and update $\pi_o$ using TD3
14: **end for**
15: **return** $\mathcal{B} = \bigcup_t (s_t, o_t, a_t, s_{t+1}, g_t)$

---

**Algorithm 3** Robust DSC Algorithm

**Inputs.** Start state $s_0$, Goal region $g$.

1: Initialize global option $o_G$ such that $\mathcal{I}_{o_G}(\cdot) = 1$
2: Initialize goal option $o_g$ such that $\beta_{o_g} = g$
3: Initialize skill chain $\mathcal{O}$ with $\{o_g\}$
4: **for** each episode **do**
5:     transitions = ROLLOUT($\mathcal{O}$)
6:     **if** $s_0 \notin \mathcal{I}_o, \forall o \in \mathcal{O}$ **then**
7:         Create new option $\omega$
8:         Add $\omega$ to skill chain $\mathcal{O}$
9:     **end if**
10: **end for**

---

**Picking a goal for option execution.** Line 4 of Algorithm 2 samples a goal from the option's termination region. To implement this sampling procedure, we consider the option's parent $\omega$ in the chain (the parent option is the one whose initiation set is being targeted by the current option $o$). We enumerate the positive examples used to train $\mathcal{I}_\omega$ and pick the goal with the highest initiation probability under the current option. This process is done iteratively backward from the goal: the first

goal is the task goal, the next one is the positive example closest (in terms of highest IVF value) to the task goal and so on.

# D    Additional Manipulation Experiments

Initiation set accuracy and true size are computed during training by performing an analogous procedure to Algorithm 1. Periodically, the manipulator was reset to each candidate start state and the initiation prediction was compared with the outcome of a policy rollout. Initiation set accuracy is visualized in Figure 8a. The methods generally converge to similar accuracy. True initiation set size is plotted in Figure 8b; size increases with optimism and correlates with success rates.

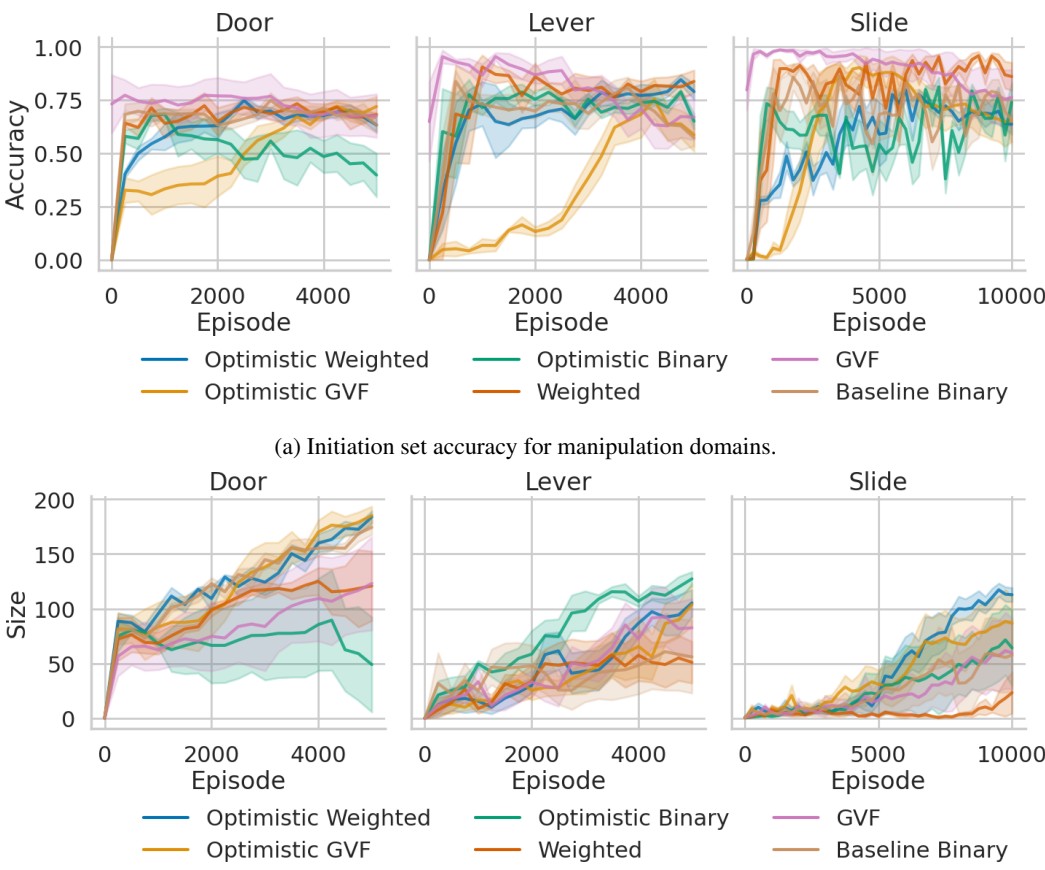

(a) Initiation set accuracy for manipulation domains.

(b) Initiation set size for manipulation domains (out of 250 start states).

Figure 8: (a) Accuracy of the learned initiation sets in the robot manipulation domains. (b) The size of the "true" initiation sets measured by performing Monte Carlo rollouts of the option policy.

# E    Ablation: Optimism

We ablate role of optimism in initiation set learning in the manipulation experiments of Section 4.2; the results are visualized in Figure 10. Count-based bonuses are computed as described in Section 3.3. The optimistic bias is critical for the success of the *Weighted* algorithm in the SLIDE task, in which only a small fraction of potential grasps are amenable to task success. For the *Binary* classification baseline, optimism dramatically deteriorates performance in the DOOR task, but yields a large improvement in the LEVER domain.

Figure 9 shows that the optimistic bonus slightly hurts in FOURROOMS, but is beneficial in MONTEZUMA'S REVENGE. This is likely because FourRooms is a much simpler problem with a smaller state-space, thereby not demanding an exploration bonus.

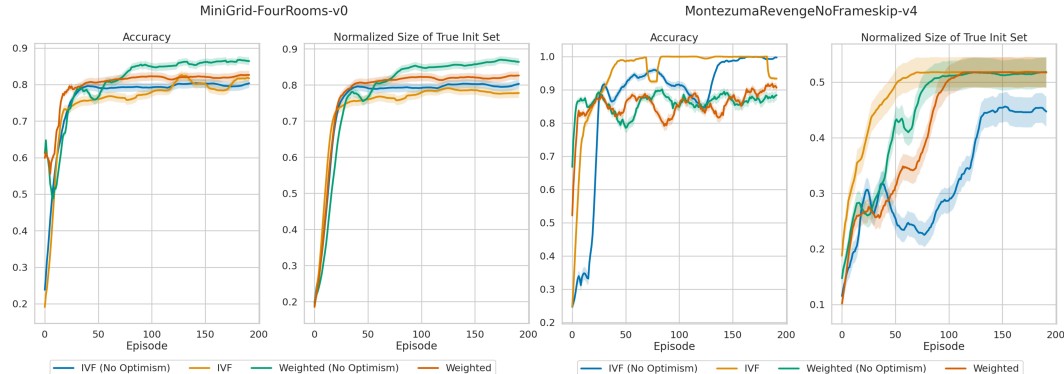

Figure 9: Optimism ablation in MINIGRID-FOURROOMS and MONTEZUMASREVENGE. All curves are averaged over all state-option pairs and 5 random seeds.

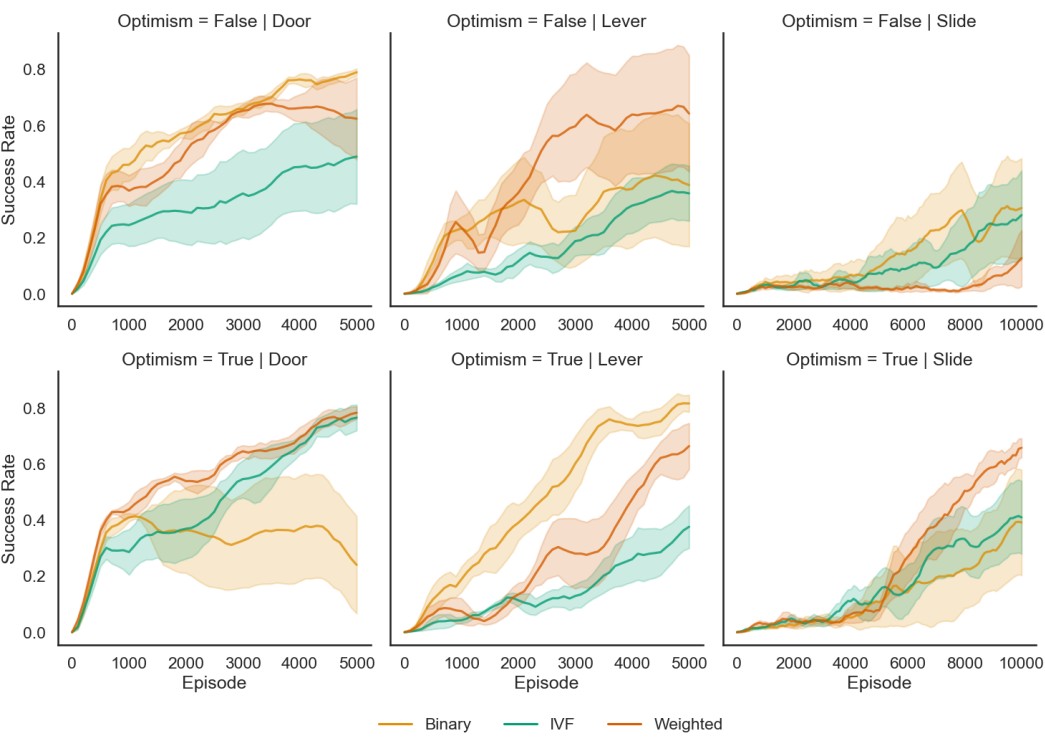

Figure 10: Optimism ablation in ROBOSUITE. Solid line represents mean success rate aggregated over four seeds; shaded region shows standard error.

## F   Initiation Sets In Four Rooms

Figure 11 shows the initiation set and IVF of the option targeting the center of the top-right room in MINIGRID-FOURROOMS. These figures show that the initiation set of the option expands over time to reflect the growing competence of the option policy.

## G   Hyperparameters

Rainbow was used for policy learning in Section 4.1, TD3 was used in the other experiments. Their hyperparameters (Tables 2 and 5) were not tuned and are either identical to the original paper implementation or borrowed from Bagaria et al. [2021a]. The bonus scale $c$ (described in Sec 3.3)

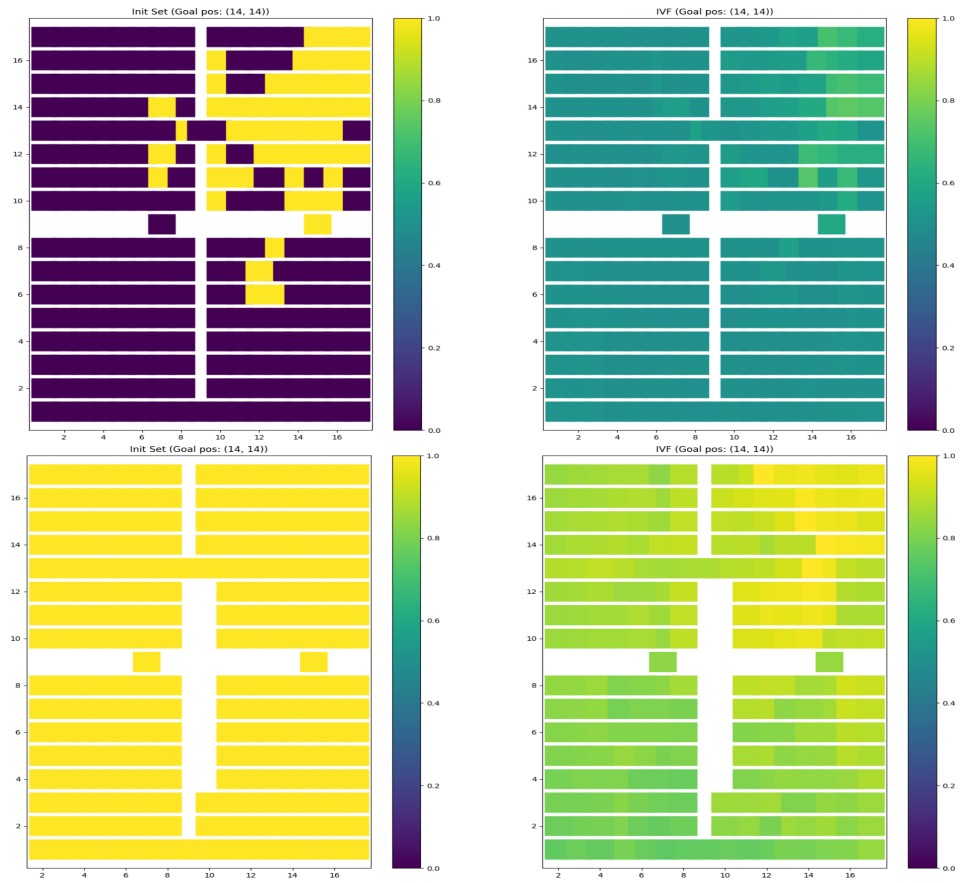

Figure 11: Initiation set *(left)* and Initiation Value Function (IVF) *(right)* of the option targeting the center of the top-right room (subgoal position is $(14, 14)$). Plots in the top row visualize the initiation set after the 1st episode of training, those in the bottom row are after 200 episodes of training.

| Env | Horizon $H_o$ |
|---|---:|
| FourRooms | 50 |
| Montezumas Revenge | 200 |
| Robosuite | 250 |
| Ant-Maze | 200 |

Table 1: Option Horizon $H_o$ for each environment.

was tuned over the set $\{0.05, 0.1, 0.25, 0.5, 1.0\}$, the best performing hyperparameters are listed in Table 3.

### G.1 Option Horizon

Options operate over a fixed time horizon $H_o$ (see Section 3.1). These values (Table 1) were picked based on overall time limit (max_steps per episode) for each domain and we did not tune them.

| Parameter | Value |
|---|---|
| Replay buffer size | $10^6$ |
| Critic Learning rate | $3 \cdot 10^{-4}$ |
| Actor Learning rate | $3 \cdot 10^{-4}$ |
| Optimizer | Adam |
| Target Update Rate | $5 \cdot 10^{-3}$ |
| Batch size | 256 |
| Iterations per time step | 1 |
| Discount Factor | 0.99 |
| Output Normalization | False |
| Policy Improvement Prior | False in ROBOSUITE; True otherwise |

Table 2: TD3 Hyperparameters for ROBOSUITE and DSC Experiments

| Method | Bonus Scale |
|---|---|
| Optimistic Binary | 0.1 |
| Optimistic GVF | 0.5 |
| Optimistic Weighted | 0.5 |

Table 3: Exploration Hyperparameters for Robosuite Experiments.

| Parameter | Value |
|---|---|
| Replay buffer size | $3 \cdot 10^5$ |
| Replay start size | 1024 |
| Learning rate | $10^{-4}$ |

Table 4: Rainbow Hyperparameters for Accuracy Experiments

| Parameter | Value |
|---|---|
| Learning rate | $10^{-4}$ |
| Optimizer | Adam |
| Replay buffer size | $10^5$ |
| Batch size | 32 |
| Threshold | 0.5 |
| Target network update rate | $5 \cdot 10^{-3}$ |

Table 5: IVF Hyperparamters

