# OpenReview forum: "Effectively Learning Initiation Sets in Hierarchical Reinforcement Learning"
_NeurIPS.cc/2023/Conference — NeurIPS 2023 poster_

### Official Review · Reviewer_uZYk · 2023-07-04

**Soundness:** 2 fair
**Presentation:** 2 fair
**Contribution:** 2 fair
**Rating:** 3
**Confidence:** 5

**Summary:**

The paper considers an HRL setting with options (including sub-goal reward functions) in which termination conditions for options are provided, and the goal is to learn the option policies and initiation sets simultaneously. The main focus is on learning of initiation sets in an online scenario where option policies are changing. The authors highlight three possible issues with learning the initiation set with classifiers in such an online setting. Firstly, the data is non-stationary because the policy keeps changing during learning. Secondly, classifiers ignore the temporal structure of the RL task. Thirdly, the difficulty of expanding the initiation set arises due to the agent not being initialized (and consequently not visiting) states outside of the initiation set.

To alleviate the first two issues, they propose combining a classifier with initiation value function (IVF) weights (expected probability of successfully completing the task from a state). They additionally propose addressing the third issue by adding an exploration bonus to IVF. The experiments compare the performance with these proposed changes against ablations without (some of) these changes. The last experiment includes a comparison against a prior method where the proposed method is used in an existing HRL algorithm.

**Strengths:**

- Originality:
  - Submission proposes an original approach for learning of initiation functions in the option setting that is considered. However, since inferring success probability is a problem that should span other parts of RL, I am not certain whether such approach isn't related to other works from different branches of RL.

- Clarity:
  - Submission is well organized (proper separation into sections and subsections etc.)
  - Data non-stationarity, pessimistic bias and temporal structure parts in section 3 are well explained

- Significance:
  - Submission tackles a task of learning initiation sets. This task is an important problem that is being researched by the HRL community

**Weaknesses:**

- Quality:
  - I am not fully convinced that the submission is technically sound. In particular:
    - I think that the problem setting described in the paper is somewhat confusing. The background section mentions SMDP and the options framework but adds internal option reward function and discount (lines 66-67). As far as I am aware, these are not standard part of the options framework. From the lines 147-150 I understand that the option policy is trained to maximize this internal reward function. This makes the problem different from the one that is considered in the original options paper [1], which relies uses a single global reward function but the paper does not elaborate on this. It is also unclear to me, what the relationship between the success condition cumulant $c_0$ and the option reward function is in section 3.1. The former is not used to learn option policy but shouldn't reaching "success" (termination states) be something that the policy is rewarded for?
    - When the IVF is introduced the timescale (discount) is set to 1 and the horizon is capped at $t=H_0$. However, there is no discussion about the effect of the $H_0$ on learning. As far as I can tell this seems to be a very important hyperparameter especially when discount is set to 1, since based on this value the size of initiation set should increase (for bigger $H_0$) or decrease (for smaller $H_0$)
    - The paper does not include qualitative visualization of the learned initiation sets and IVF in the easy environment (four rooms). I think that such visualization should be in the paper to evaluate and justify proposed changes and help to understand their effect on learned initiation sets and IVF.
    - It is unclear from the experiments whether (and in which cases) proposed adjustments help. The plots (in particular Figure 2), seem inconclusive to me.
    - Authors do not discuss the limitations/weaknesses of their method
    - In Figure 5., the prior work baseline seems to perform very poorly. While I understand that the setting was slightly adjusted, I am not sure whether the baseline was properly tuned.

- Clarity:
  - The comparisons are sometimes not well explained (what are GVF baseline in Fig. 2?)

[1] Sutton, R. S., Precup, D., & Singh, S. (1999). Between MDPs and semi-MDPs: A Framework for Temporal Abstraction in Reinforcement Learning

**Questions:**

- In the abstract you mention that option policy and initiation set must be learned simultaneously? Could you elaborate on why this is the case?
- In line 92 you mention that "initiation function is used in the context of primitive actions". What does context of primitive actions mean here?
- Regarding data non-stationarity part in section 3. Wouldn't it be possible to discard old training samples based on the probability of current policy executing old trajectories?
- Is it possible to use IVF with thresholding directly instead of learning a weighted classifier? If yes, why haven't you included this baseline in the paper?
- Why does the option use internal reward function instead of a global one?
- Shouldn't reaching "success" (termination states) be something that the option policy is rewarded for?
- If the option maximizes internal reward function, is the problem of finding policy $\pi_0$ pretty much an MDP $<S,A,R_0, T, \gamma>$?

Comments:
- line 70 - you cite Feudal RL with "usually assume that all options apply everywhere" afaik, this approach does not use options and in fact was proposed before options
- line 109 - It is probably better to cite the full paper Options of Interest: Temporal Abstraction with Interest Functions (Khetarpal et al. 2020) from AAAI 2020 as a reference to IoC



**Limitations:**

Authors do not sufficiently discuss the limitations/weaknesses of their method

---

> ### Author Rebuttal · Authors · 2023-08-10
>
> We are happy that you found our method to be original and the problem setting to be important. We hope that we can clarify some misunderstandings and convince you about the value our proposals to the HRL community.
>
> ## Weaknesses
>
> > Inferring success probability is a problem that should span other parts of RL
>
> We agree the problem is very general and we hope that our solution is also adopted by other parts of RL. However, we are not sure how that is a weakness of our work.
>
> > The background section mentions SMDP and the options framework but adds internal option reward function and discount, which makes the problem different from the one in the original options paper, which relies on a single global reward function.
>
> It is very common in HRL for options to have their own internal reward functions [1]; this was discussed in the HRL survey paper [3] (whose formalism we adopt in our Background section) and has been the case in option discovery research for several decades [2]. We can add more formalism around this in the background section if you want, but we hope that this is not a reason for rejection.
>
> > What is the relationship between the success cumulant and the option reward function in section 3.1?
>
> The option reward function is used to learn the option policy--it depends on the environment and is designed to enable sample efficient policy learning. For e.g, in Robosuite, the reward function is dense and depends on the configuration of objects in the scene. However, the initiation cumulant is fixed and environment agnostic---it is always $+1$ for reaching the goal and $0$ otherwise. However, this cumulant should not always be used for policy learning; for e.g, you might want negative rewards for states to avoid or shaped rewards for sample efficiency.
>
> > When the IVF is introduced the discount is set to 1 and the horizon is capped at H0. However, there is no discussion about the effect of the horizon on learning.
>
> In our experiments, we treated the horizon as a hyperparameter but forgot to include them in the appendix. Here are the values: MiniGrid $50$, MontezumaRevenge $200$, RoboSuite $250$, AntMaze $200$ (all methods used the same horizon). These numbers were picked based on the overall time limit (max_steps/episode) for each domain and we did not tune them. We apologize for the omission and will include this information in the camera-ready if the paper is accepted.
>
> Based on your suggestion, **we ran an experiment in which we swept over the option horizon $H_o$** in MiniGrid-FourRooms and recorded the accuracy and size of the learned initiation sets; the results are included in the accompanying PDF. Option horizon $H_o$ has a predictable effect on accuracy: as long as $H_o$ is not too small ($\geq 40$ timesteps), accuracy is high. Your intuition about the relationship between horizon and initiation set size is *mostly* correct: $H_o$ sets an upper bound on the size of the initiation set, the final size is also dependent on the competence of the learned policy and the connectivity of the underlying MDP.
>
> > The paper does not include qualitative visualization of the learned initiation sets and IVF in four rooms
>
> Thank you for your suggestion, **we visualized the learned classifier and IVF** for MiniGrid-FourRooms; see Fig A in the accompanying PDF.
>
> > The plots (in particular Figure 2), seem inconclusive to me.
>
> Figure 2 shows that our method achieves \~$25$-$30$% increase in accuracy in MiniGrid and \~$50$% increase in accuracy in Montezuma’s Revenge; these results suggest that our techniques **conclusively** and significantly outperform the baseline approach to learning initiation sets. To be clear, the line labeled “Baseline Binary” is the only one that is not our contribution and corresponds to the method used in almost all prior work; the other three lines are novel to our paper.
>
> > (Fig 5) I am not sure whether the baseline was properly tuned.
>
> We used the author implementation of baseline DSC. We reproduced their reported results when the initiation was learned over the $(x, y)$ location of the ant, but the performance drops dramatically when the full state is used for initiation learning. We tried different values of gestation period ${5, 10}$ and learning rates {1e-3, 1e-4, 1e-5}; none of these settings could solve AntMediumMaze. Our algorithms (Weighted Clf and Initiation GVF) were also implemented in the author codebase for an apples-to-apples comparison. The DSC codebase is open-source, so if you want to try the experiment yourself, we are happy to share the run commands and configs we used.
>
> ## Questions
>
> > Why should the option policy and initiation set be simultaneously learned?
>
> The initiation set evaluates the reachability of the current policy, so as the policy changes in the continual learning setting, so should the initiation set, i.e, they depend on each other.
>
> > Is it possible to use IVF with thresholding directly instead of learning a weighted classifier? If yes, why haven't you included this baseline in the paper?
>
> We *do* use the IVF by itself – since it always outputs values between 0 and 1, it is much easier to threshold than the value function used for action-selection (see Section 3.1). We should clarify that the thresholded IVF is *not* a baseline, it is one of the two algorithms we propose. That line was mistakenly labeled as “GVF” in Fig 2, but it should read “IVF”.
>
> > If the option maximizes internal reward function, is the problem of finding policy pretty much an MDP?
>
> Yes, that is correct! In fact, M White et al 2017 formalized options as RL subtasks.
>
> ## References
>
> [1] Precup, Temporal abstraction in RL, PhD Thesis (2000).
>
> [2] McGovern & Barto. Automatic discovery of subgoals in reinforcement learning using diverse density (2001).
>
> [3] Barto & Mahadevan. Recent Advances in Hierarchical Reinforcement Learning (2003).
>
> [4] Sutton et al. Reward-respecting subtasks for model-based reinforcement learning (2022).

---

> > ### Comment · Reviewer_uZYk · 2023-08-16
> > **Response to rebuttal**
> >
> > Thank you for your response.
> >
> > - **We agree the problem is very general and we hope that our solution is also adopted by other parts of RL. However, we are not sure how that is a weakness of our work.**
> >
> >   In the setting that is considered (terminations are given, internal reward function), the problem becomes learning a policy in an MDP while also inferring whether a policy can reach certain state in a set time horizon. While here it is mostly considered in hierarchical setting, it is quite likely that similar setting may arise non-HRL areas of RL. My comment was not in the weaknesses section because I do originality of the work to be a weakness. I do think that there is an original contribution. This comment reflected my uncertainty about possible connection to prior related work outside of HRL space.
> >
> > - **It is very common in HRL for options to have their own internal reward functions...**
> >
> >   Thank you for the clarification. Could you please incorporate these references into the background section in the next revision?
> >
> > - **In our experiments, we treated the horizon as a hyperparameter but forgot to ...**
> >
> >   I appreciate additional results and have some comentes/questions:
> >    - When you calculate accuracy shouldn't Figure A (c) result in 100%? I suppose that you get <100% for plots because you consider successes from samples for this metric but wouldn't it better to evaluate accuracy by comparing (c) with true initiation set (tresholded true IVF or something similar)? Could you maybe elaborate on how accuracy is calculated?
> >    - I don't think that the setting in which H=50 is the most relevant because (as far as I can tell) simply using trivial initalization set I=S would lead to the best performance (100% accuracy and set size). This is because it should take less than 50 steps from each S to reach the goal. Is there something I am misunderstanding? I think that using lower H where there is a region where options should not be initialized is much more interesting since it imitates real usecase and requires your method to find I that are different from S.
> >    - Are these plots in Figure A with H=50?
> >    - Why does the performance of the algorithm drop with lower H? As far as I can tell, there should be no reason why the accuracy should drop if the classifier can predict states that can reach the goal correctly. Having plots similar to Fig. A with H<50 would probably help explain this?
> >    - I think that it would be useful to have similar experiments (with changed H) for other settings to see the effect of this (important) hyperparameter
> >    - Is the goal reachable from all initial states within H steps in montezuma's revenge?
> >
> > - **We used the author implementation of baseline DSC. We reproduced their reported results when the initiation was learned over the
> >  location of the ant, but the performance drops dramatically when the full state is used for initiation learning. ...**
> >   - I think that it would be best if the results on the original domain would be included in the paper (together with the new domain) for a fairer comparison since tuned hyperparameters are available for that domain unless there is an explicit argument for why DSC baseline would not be able to handle the new setting.

---

> > > ### Author Response · Authors · 2023-08-18
> > > **Further clarifications**
> > >
> > > Thank you for continuing to engage with our work, we appreciate it!
> > >
> > > > Could you please incorporate these references into the background section?
> > >
> > > Absolutely!
> > >
> > > > Could you maybe elaborate on how accuracy is calculated?
> > >
> > > At each state $s\in\mathcal{S}_0$, we record the initiation decision made by the learning algorithm as $\hat{\mathcal{I}}_o(s;\theta)\in\\{0, 1\\}$. We then execute the option policy $\pi_o$ from that state and record whether or not the agent reached the option's subgoal as $Y_s\in\\{0,1\\}$. Accuracy at state $s$, for option $o$ is then given by the indicator $1(\hat{\mathcal{I}}_o(s;\theta)=Y_s)$. This process is repeated several times for all options $o\in\mathcal{O}$ and start states $s\in\mathcal{S}_0$. Detailed discussion in Section 4.1 and Appendix B.1.
> > >
> > > > Wouldn't it be better to evaluate accuracy by comparing A(c) with the true initiation set?
> > >
> > > Yes, evaluation would be easier if we had access to the true initiation set, but we don't; we only have samples from it. In FourRooms, the ground-truth initiation set might be true everywhere at the *end of training*, but we cannot always treat I=S as ground-truth because:
> > > - We do not have similar prior knowledge in other domains where the observation space is high-dimensional and harder to reason about.
> > > - The ground-truth depends on the reachability of the *current* policy: if the policy is newly initialized, its ground-truth initiation set is not true everywhere.
> > >
> > > > The trivial initialization set I=S would lead to the best performance (100% accuracy and set size). Is there something I am misunderstanding?
> > >
> > > Yes, you are confusing the initiation set of the *current* policy with that of the *optimal* policy: we approximate the former by framing the problem as policy evaluation. So setting $\mathcal{I}=\mathcal{S}$ in FourRooms would register low accuracy most of the time because the policy would be unable to reach the goal within H steps, i.e, the ground-truth initiation set would not be true everywhere. Similarly, the size of the true init set would not be 100% because in early stages of learning, the policy does not have 100% competence. The best we can hope to do with non-stationary policies is to *track* their initiation set, which is what our algorithms attempt to do.
> > >
> > > > Are these plots in Figure A with H=50?
> > >
> > > Yes, we used $H=50$ in Figure A because that was the hyperparameter setting we used in the main paper.
> > >
> > > > Why does the performance of the algorithm drop with lower H?
> > >
> > > For very small values of H, the initiation classifier decides that only states that are right next to the goal (say 2 steps away) should be inside the initiation set. However, the stochastic policy is still able to sometimes reach the goal from states that are outside the learned classifier (say 4 steps away), just not reliably. This is why the accuracy is both low and high variance for very small values of H.
> > >
> > > We have plots similar to Figure A for different values of H, but could not include in the rebuttal PDF because of space constraints. We are happy to include it in the main paper.
> > >
> > > > It would be useful to have similar experiments (with changed H)
> > >
> > > We will be sure to include experiments about the option horizon in the main paper, thank you!
> > >
> > > > Is the goal reachable from all initial states within H steps in Montezuma's Revenge?
> > >
> > > A horizon of H = 200 steps is sufficient for an optimal policy to reach the goal from all initial states in Montezuma’s Revenge. However, with finite data, training time and function approximation, we unsurprisingly do not recover the optimal policy. Furthermore, the initiation set captures the set of states from which the policy can succeed *with high probability*. So, even if the policy might *sometimes* reach the goal from some states, they would not be in the initiation set because of our insistence on reliability.
> > >
> > > > [DSC] I think that it would be best if the results on the original domain would be included in the paper
> > >
> > > Baseline DSC uses privileged information about which state variables are necessary for initiation set learning: they use the (x, y) location of the ant; we do not. Comparing our algorithm to a version of DSC which has access to privileged information would not be an apples-to-apples comparison, which is why we did not include that result in Figure 5.
> > >
> > > Figure 6 in [1] shows that the baseline DSC (with privileged information) gets a success rate of ~50%, our algorithm gets a success rate >80% in roughly the same number of episodes. So, not only does our method remove the need for privileged information about which state variables are relevant for initiation learning, it also gets higher reward. If you want, we can include the baseline's final reported result as a dotted line in our plot.
> > >
> > > Thank you once again for your time, we appreciate it! **Are there any other questions you want answered during the discussion period to consider raising your score?**
> > >
> > > [1] Robustly learning composable options in deep RL, IJCAI 2021.

---

> > > > ### Comment · Reviewer_uZYk · 2023-08-21
> > > > **Response**
> > > >
> > > > Thank you for your response.
> > > >
> > > > - *True initiation set* - I agree that in most tasks that are continuous/high-dimensional getting true initiation set would be impossible. However, as far as I can tell this should be easy in FourRooms because of small finite state/action space and should be a simple policy evaluation using cumulant $c$. I would argue that this may be a better way than sampling for toy task like FourRooms.
> > > >  - *drop with lower H* - I understand that there might be regions where there is non-binary success rate. However, as far as I can tell these should be relatively small, for example for H=10 everything that is further than 10 steps from 14,14 should be predicted as 0 which should be two left rooms (-1 cell) and majority of the bottom right room. Predicting only these states correctly should lead to more than 50%-60% accuracy. Also, what I am a bit puzzled by is the performance of bigger Hs. I am not sure why 60, 80 and 90 are worse than 50 and why 80 is worse than 60 and 90. I don't think that bigger H should have too much of an effect on performance in this case.
> > > >   - *confusing the initiation set of the current policy with that of the optimal policy* - I understand the difference that you are pointing out. I agree that the main difference is that you are tracking $I$ over time and having access to intermediate which should be beneficial for the learning of options during training. I still think that settings where final $I$ would be different from $S$ would be more interesting because it would imitate the setting in which options are not executable from everywhere (after training) which is one of the main benefits of using initiation sets.
> > > > - *DSC* - I understand that you are using full state instead of x,y. However, in my opinion, having a separate plot with the *old setting (x,y) that includes your method and DSC* either in the main paper or appendix would help to increase the confidence in results shown in Figure 5.
> > > >
> > > > At the moment, after seeing the results of additional experiments with different H values in FourRooms, I still have some doubts about the correctness of the method and experiments. However, I will consider adjusting my score after the discussion with other reviewers.

---

> > > > > ### Author Response · Authors · 2023-08-22
> > > > > **Small Clarifications**
> > > > >
> > > > > Thank you for your continued engagement with our work, we really appreciate it!
> > > > >
> > > > > > I still think that settings where final would be different from would be more interesting
> > > > >
> > > > > All problems other than FourRooms (Montezuma's Revenge, Robosuite, Ant Maze) have initiation sets that are smaller than the state-space, even at the end of training. For e.g, in our Robosuite experiments, the robot cannot make a good grasp from most states (see Figure 4).
> > > > >
> > > > > > A separate plot with the old setting (x,y) that includes your method and DSC either in the main paper or appendix would help to increase the confidence in results shown in Figure 5.
> > > > >
> > > > > Okay, we will include these results in the camera-ready, thank you!
> > > > >
> > > > > > Discussion about option horizon H_o
> > > > >
> > > > > Zooming out, we are discussing the effects of horizon on the RL problem. Your expectations/intuitions are absolutely correct for tabular RL; however, when combined with function approximation, horizon/discounting have unintuitive effects on training dynamics. This phenomenon is well documented, although not very well understood. For interesting examples, see *On Inductive Biases in Deep Reinforcement Learning* by Hessel et al and *Time Limits in Reinforcement Learning* by Pardo et al.
> > > > >
> > > > > In short, we agree that option horizon is an important hyperparameter. But, the unintuitive effect of H_o on the final value function is not specific to our algorithm, it generally applies to all of deep RL and we hope that it is not a reason for rejecting our paper. We are grateful that you pointed this out, we will certainly include a detailed analysis in the camera-ready.

---

### Official Review · Reviewer_y3Ji · 2023-07-04

**Soundness:** 4 excellent
**Presentation:** 3 good
**Contribution:** 4 excellent
**Rating:** 7
**Confidence:** 4

**Summary:**

The paper presents an approach for efficiently learning initiation sets of options for approaches that automatically learns temporal options for hierarchical planning. The paper introduces the concept of initiation value function (a probability measure for successful option executions for a given imitation set) and uses IVF to learn initiation sets of options while option policies are also changing. Lastly, the approach learns a neural network for predicting the probability of successful option execution using a classifier. The authors evaluate their approach in variety of settings including discrete spaces (minigrid settings) as well as continuous robotics settings.

**Strengths:**

- The paper is well written. Details are highlighted nicely.

- Authors have thoroughly presented related work and mentioned differences with existing works.

- The experiments are thorough. Sufficient details are provided for experimental setup.

- The paper successfully motivates the readers about the technical challenges in learning initiation sets of the option using the traditional binary classifier based approach and provides clear references to what part of their approach tackles each problem. This makes understanding the paper and following the approach lot easier.

**Weaknesses:**

The only weakness I see of the paper is lack of clarity on how the approach is used with an option discovery approach. The given approach requires a policy to compute IVF and learn a neural network. However, for problems that require option discovery, do not have these options along with a learned policy. It is unclear from the paper how was this made possible without having a fixed policy.

It would be helpful to highlight how the approach was configured to be used with DSC.

**Questions:**

Please refer to the weakness mentioned above.

**Limitations:**

The authors do not explicitly address the limitations of the paper.

In my opinion the rest of the paper (apart from not explicitly highlighting the limitations and the weakness mentioned about) is written nicely.

---

> ### Author Rebuttal · Authors · 2023-08-09
>
> We are delighted that you thought that our paper was well written, thorough and easy to follow!
>
> We understand your desire for more details about how our methods are incorporated into the option discovery algorithm, DSC. Section B.3 (including Algorithms 2 and 3) in the Appendix provides more details about the DSC algorithm and the experimental setup. A lot of these details are part of the DSC algorithm, which is not our contribution, so we deferred the discussion to the Appendix. However, we are happy to add some more details to the main paper so that Section 4.3 is easier to follow.
>
> Based on your suggestion, we will also add a discussion about the limitations of our method to the camera-ready.
>
> Thank you once again for your thoughtful review!

---

> > ### Comment · Reviewer_y3Ji · 2023-08-18
> >
> > Thanks for the response and acknowledging the question.

---

### Official Review · Reviewer_pGKy · 2023-07-06

**Soundness:** 3 good
**Presentation:** 2 fair
**Contribution:** 3 good
**Rating:** 5
**Confidence:** 2

**Summary:**

This paper addresses the problem of learning initiation sets in hierarchical reinforcement learning. Initiation sets define the states from which an option can be executed successfully. However, learning initiation sets is challenging because they depend on the option policy, which changes as the agent learns. Previous approaches suffer from data non-stationarity, pessimistic bias, and a lack of exploitation of temporal structure. To address these issues, the authors propose the use of the Initiation Value Function (IVF) as a predictive value function that estimates the probability of option success. The proposed method is evaluated in various domains, including grid worlds, robot manipulation, and maze navigation, and shows improved performance compared to existing methods.

**Strengths:**

1) Novelty: the paper presents a novel method for learning initiation sets from off-policy trajectories by incorporating the temporal structure of the MDP. It also introduces an original technique to mix leverage the strengh of classification togfether with TD learning.

2) The evaluation of the proposed method is comprehensive, with separate experiments conducted to systematically test and validate each claim.

3) The performance of the proposed method, particularly when incorporating the bonus correction, show significant improvements compared to existing methods.

**Weaknesses:**

1) There are certain parts of the paper that lack clarity, such as the algorithm for selecting the goal for the DSC method. Additionally, Figure 3 is challenging to interpret due to the excessive use of similar colors.

2) The paper primarily focuses on the DSC algorithm and does not explore the application of the proposed method to other option frameworks where termination states are also learned simultaneously. Expanding the evaluation to different option frameworks would have strengthened the paper's overall contribution.

**Questions:**

1) Is it possible to use TD learning with cross-entropy loss from the logits instead of regression? Would this approach outperform TD learning with regression?

2) Regarding the "Initiation Set Accuracy" plot (Fig 2), it would be valuable to have a similar plot that compares the different methods while choosing a single option policy learning shared by all methods, focusing solely on the variation in the initiation set learning. The current plot, although informative, compounds the evaluation of initiation set learning with the resulting option policy learning.

3) The abstract mentions that the termination condition is typically identified first. However, the algorithm could potentially work even when the termination policy changes. Could you provide insights into the performance of the method if the termination state is learned in parallel? Alternatively, how restrictive is the assumption?

**Limitations:**

see weaknesses

---

> ### Author Rebuttal · Authors · 2023-08-09
>
> Thank you for your thoughtful review and suggestions. We are glad that you found our work to be novel and our experimental approach to be comprehensive and systematic!
>
> > There are certain parts of the paper that lack clarity, such as the algorithm for selecting the goal for the DSC method.
>
> The algorithm for selecting goals for DSC is discussed in Section B.3 of the Appendix. Additionally, [1] discuss the nuances of goal-selection in DSC. Briefly, the subgoal state for the current option is the highest probability (IVF prediction from the current state $\mathcal{V}_o(s_t)$) positive example used to train the parent option’s initiation set.
>
> > Figure 3 is challenging to interpret due to the excessive use of similar colors.
>
> You are right, thank you, we will improve the presentation for the camera-ready if the paper is accepted.
>
> > Expanding the evaluation to different option frameworks would have strengthened the paper's overall contribution.
>
> We chose DSC because of its explicit dependence on initiation set learning. Having said that, the primary focus of our experiments is evaluating how accurate the initiation sets are, and we added a secondary experiment to show the effect of that on downstream learning. There are many option discovery methods and we expect this approach will help all of them similarly.
>
> > Is it possible to use TD learning with cross-entropy loss from the logits instead of regression? Would this approach outperform TD learning with regression?
>
> Yes, that is a good suggestion: using distributional RL might further improve initiation set learning. We used the traditional TD error in this paper because of its simplicity and because distributional RL is not the focus of our paper.
>
> > The current initiation set accuracy plot (fig 2), although informative, compounds the evaluation of initiation set learning with the resulting option policy learning.
>
> Initiation set learning is extensively studied and well understood when the option policy is fixed; in that setting, it is just binary classification. While we agree that learning initiation sets in tandem with policies complicates evaluation, that is exactly the focus of our paper. We introduced the "initiation set size" metric alongside accuracy specifically to understand the interplay between initiation set learning and policy improvement, which naturally arises during continual/online reinforcement learning.
>
> > Could you provide insights into the performance of the method if the termination state is learned in parallel? Alternatively, how restrictive is the assumption?
>
> DSC uses goal-conditioned policies to deal with non-stationary termination/subgoal regions, and our proposed methods seem to work very well with DSC, so in that sense we already have an experiment that does what you are asking for (Figures 5 and 7). Other than that, if the termination condition of the option changes, its policy will likely degrade; since our initiation value function (IVF) is learning using off-policy policy evaluation, it will adaptively lower the policy’s probability of success. Over time, as the policy becomes more competent at the new subgoal, the IVF’s predictions will also increase. In other words, as long as the policy learning adapts to the non-stationarity, we fully expect our initiation set learning to also adapt.
>
> Thanks again for your thoughtful reviews! If you have any questions/concerns, please let us know.
>
> [1] Bagaria, Akhil, et al. "Robustly learning composable options in deep reinforcement learning." Proceedings of the 30th International Joint Conference on Artificial Intelligence. 2021.

---

> > ### Author Response · Authors · 2023-08-18
> > **Any additional questions or concerns?**
> >
> > We hope that we have been able to resolve most of your concerns via our rebuttal. Are there any other questions/concerns you want addressed during the discussion period to consider raising your score?

---

### Official Review · Reviewer_zsHA · 2023-07-27

**Soundness:** 3 good
**Presentation:** 3 good
**Contribution:** 2 fair
**Rating:** 3
**Confidence:** 3

**Summary:**

This paper studies the problem of learning initiation sets, which are important in hierarchical reinforcement learning for indicating where a policy will succeed, and identifies three main challenges that arise with existing methods to learn these sets. These three challenges are data non-stationarity, temporal credit assignment, and pessimism. The authors introduce the Initiation Value Function (IVF), which predicts the probability that an option will succeed from that state, which is learned using off-policy policy evaluation and adapts as the policy improves. IVF is effective when used as a input to a weighted classifier and additionally includes states for which the option policy is likely to improve in order to address pessimistic bias. Experiments in Minigrid, Montezuma's Revenge, Robosuite, and a maze navigation problem are shown, and in these environments, the method improves upon the baseline of using a General Value Function.

**Strengths:**

- The paper is generally well-written and has empirical evaluations on a wide range of domains.
- The paper brings up some good discussion on three challenges to address for learning initiation sets in hierarchical RL.

**Weaknesses:**

-The paper involves adding a lot of complicated tricks into a hierarchical method, and it's unclear if they're worth it for the often small improvements. I'm not sure if people will want to adopt this method. It's also unclear which of the challenges is most important, and which of the tricks added is most important. Can you add some more analysis/discussion on this?
- It's a bit strange how the authors emphasize the important of exploiting temporal structure to learn IVF, in order to be more effective than a classifier, but then just uses IVF to weight a classifier. Seems like a lot of effort to learn a value function just to weight the classifier. It's not clear if the temporal aspect is actually meaningful in the classifier. For example, how does this compare to just using binary classifier that has mixup regularization or other types of heavier regularization along with the optimism?

**Questions:**

See weaknesses. Additionally:
-In the Figure captions (or elsewhere), can you clearly define what the each method is? Figure 2, for example, does not have a method called "IVF" so it's unclear which is yours. Also, it'd be helpful to add y-axis labels for all the plots.

**Limitations:**

The authors have not addressed limitations. It would be helpful to include discussions of limitations.

---

> ### Author Rebuttal · Authors · 2023-08-10
>
> We are glad that you found our paper to be well written and our empirical evaluations to be diverse. We hope that we can change your mind about the potential value of our paper to the HRL community.
>
> > In the Figure 2, can you clearly define what each method is?
>
> We accidentally labeled the IVF approach as “GVF”, we will fix that for the camera-ready if the paper is accepted.
>
> Here is a brief description of the different lines in Figure 2:
> 1. **Baseline Binary.** Binary classifier used to learn the initiation set. This is what is used in essentially all prior work and is the only baseline method in this figure.
> 2. **GVF.** Threshold the Initiation Value Function (IVF).
> 3. **Optimistic GVF.** Threshold the sum of IVF and an exploration bonus.
> 4. **Weighted.** This is a weighted binary classifier where the weights are assigned by the IVF.
>
> Again, (1) is a baseline; (2), (3), and (4) are our contributions.
>
> > The paper involves complicated tricks with small improvements. It is unclear which of the tricks added is most important.
>
> We are not sure what the reviewer means here: our proposed methods are simple and they lead to substantial performance increase. Figure 2 aims to ablate the different contributions, are there specific “tricks” whose contribution you are unsure about?
>
> Furthermore, we did not randomly add tricks to see if they increase accuracy/reward. Instead, we are introduced modifications specifically designed to address an important, known issue in HRL: the collapse of initiation sets [1, 2].
>
> > It's a bit strange how the authors emphasize the importance of exploiting temporal structure to learn IVF, but then just uses IVF to weight a classifier.
>
> First, we show that thresholding the IVF learns better initiation sets than the baseline binary classification approach, so we do not *just* use the IVF to learn a classifier. Our second contribution is the IVF-weighted classifier, which adds a temporal dynamics-aware target to combine the strengths of the value function and the classification approaches.
>
> In *Montezuma’s Revenge*, the pure IVF and weighted classifier approaches increase accuracy by $50$% and $40$% respectively over binary classification; in *AntMediumMaze*, they are the difference between solving the task reliably ($>80$% success rate) and not solving the task at all ($0$% success rate), so our approaches are certainly worth the effort.
>
> Mixup regularization (or anything similar) is a good suggestion, but it cannot learn temporal structure, TD learning (or temporal bootstrapping in general) is required for that. To show the benefit of using TD for learning the IVF, **we have included a new experiment in the accompanying rebuttal PDF**: Figure B shows that even when a trajectory does not reach the goal, TD unsurprisingly assigns positive credit to states that are near the goal, thereby aiding more sample efficient learning.
>
> ## References:
> [1] Harb, Jean et al. When waiting is not an option: Learning options with a deliberation cost. In Thirty-Second AAAI Conference on Artificial Intelligence, 2018.
>
> [2] Bagaria, Akhil, et al. "Robustly learning composable options in deep reinforcement learning." Proceedings of the 30th International Joint Conference on Artificial Intelligence. 2021.

---

> > ### Author Response · Authors · 2023-08-18
> > **Any additional questions or concerns?**
> >
> > It seems to us that your negative review was mostly because of:
> > - confusion regarding what the different lines in Figure 2 meant
> > - lack of clarity about the importance of using TD when learning the IVF
> > - missing discussion of the limitations of our work
> >
> > We hope that we have been able to resolve these concerns in our rebuttal. If you have any lingering questions, we would be happy to address them in the remaining time. Thanks again for your time and engagement!

---

### Official Review · Reviewer_RLJn · 2023-08-02

**Soundness:** 3 good
**Presentation:** 3 good
**Contribution:** 3 good
**Rating:** 7
**Confidence:** 4

**Summary:**

The paper discusses three difficulties in effectively learning option initiation sets in reinforcement learning --- these are non-stationarity, temporal credit assignment, and pessimism --- and proposes a method that addressed these difficulties. The proposed method is evaluated empirically in a variety of environments. The results are positive.

**Strengths:**

This paper makes a good contribution on a topic of interest. It is clear and well structured. The algorithmic approach is well motivated and sound. Experimental analysis is conducted on a range of environments and, importantly, includes the use of the proposed method within a specific skill discovery algorithm, with positive results. Overall, I believe the paper is a useful addition to the literature.

**Weaknesses:**

It would be useful to see a discussion on the computational cost of the proposed approach.

The experimental evaluation can be broadened to include a detailed analysis of the utility of the weighting function (equation 1) and the discussion could include whether any other alternatives would be sensible. In addition, it would be useful to see experimental results with the proposed approach in the context of skill discovery methods other than deep skill chaining.

Additional Comments:

In the abstract, and later in the paper, the authors state that they identifiy and address three difficulties in effectively learning option initiation sets: non-stationarity, pessimism, and temporal credit assignment. In the introduction, I do not see any reference to temporal credit assignment.

In line 201, the bonus reward is the change in the value function. This was explored by Simsek & Barto (ICML, 2006, An Intrinsic Reward Mechanism for Efficient Exploration) earlier than the papers cited here.

In figure 2, the second plot from the right: either the plot title or the label of the vertical axis is wrong. The title is "size of the true initiation set" but the axis labels range from 0.2 to 0.9. The same holds for the last plot in the figure.

The writing is generally well structured and clear but at times repetitive. Reducing the repetition would allow the authors to include in the main paper (rather than the supplementary material) the results on initiation set accuracy and size in the robot manipulation environments, which are informative. If space cannot be found in the main paper for these plots, the results can still be summarised and discussed in the main paper. These plots show key performance variables.

**Questions:**

Have you considered alternatives to the weighting function in Equation 1?

Note: I have read the rebuttal. Thanks to the authors for answering my question and their additional comments.

**Limitations:**

I do not see a discussion of limitations.

---

> ### Author Rebuttal · Authors · 2023-08-09
>
> Thank you for the positive review. We are glad that you found our paper to be clear and our approach and experiments to be sound.
>
> > It would be useful to see a discussion on the computational cost of the proposed approach.
>
> The computational cost of the IVF approach is very similar to that of the baseline approach of training a binary classifier. The weighted classifier is more expensive since the IVF is used to weigh each of the training examples for the classifier. However, it is worth noting that our new approaches lead to faster experiments in Section 4.3: when the agent reaches the goal in a given episode, that episode is shorter than 1000 steps, so the overall experiment finishes in a lot fewer environment steps than the baseline. In other words, although we add some compute, we drop sample complexity dramatically.
>
> > Have you considered alternatives to the weighting function in Equation 1?
>
> Eq 1 was the simplest way to incorporate the IVF into a weighting function. Experimentally, we verified that the weighting function adapted to a changing policy over time. Finally the weighted classifier was accurate and performed well in the skill-discovery setting, so we did not feel the need to try more complex weighting functions.
>
> > it would be useful to see experimental results with the proposed approach in the context of skill discovery methods other than deep skill chaining (DSC).
>
> We chose DSC because (a) it shows strong performance in sparse-reward continuous problems and (b) its performance is sensitive to initiation set learning. It is possible that other skill-discovery algorithms would also benefit from better initiation set learning, but we leave that analysis to future work.
>
> > In figure 2, the second plot from the right: either the plot title or the label of the vertical axis is wrong.
>
> We apologize for the confusing axis label: the y-axis is the *normalized* size of the initiation set---it is the fraction of states inside the ground-truth initiation set (described in Section 4.1), so it is a real number between 0 and 1. We will change the axis labels to make this more clear.
>
> > This was explored by Simsek & Barto (ICML, 2006, An Intrinsic Reward Mechanism for Efficient Exploration) earlier than the papers cited here.
>
> You are right, thank you, we will certainly cite this paper in the camera-ready.
>
> Thank you again for the positive review and thoughtful suggestions.

---

### Author Rebuttal · Authors · 2023-08-10

We are grateful to all the reviewers for their thoughtful reviews and constructive feedback. We are happy to see that all reviewers found our work to be novel and our paper to be well written and clear in its presentation. We hope that our rebuttal helps resolve misconceptions and remaining reservations about our work.

## Misconceptions

Some reviewers seemed to think that the GVF approach in Fig 2 was distinct from the IVF approach in Section 3.1; they are the same – this approach thresholds our Initiation Value Function (IVF). Furthermore, thresholding the IVF is not a baseline, it is one of our two contributions (in addition to the weighted classifier). This is because the prevalent way to use value functions as initiation sets is to threshold the option value function that is also used for action-selection. But as we discussed in Sections 2 and 3.1, that is not a good idea because it either leads to difficulties in policy learning or involves complicated and fragile normalization tricks.

## Common Concern

A common cause for concern was our use of DSC as a skill-discovery algorithm in Section 4.3. We used DSC for two reasons:
- It is a recent option discovery algorithm that performs well in sparse reward control problems and DSC learns options whose subgoals are initiation sets of other options, i.e, its performance is heavily dependent on learning good initiation sets. Because learning high quality initiation sets is central to DSC, it makes for a good test-case for our proposed ideas.
- Furthermore, the experiments preceding that section are meant to evaluate initiation set accuracy directly. This one is to demonstrate initiation set accuracy's effect on downstream learning. Of course that depends on the skill discovery algorithm, of which there are many, so we picked a representative and recent one.

## Limitations of our Work

Reviewers suggested that we discuss limitations of our method: our proposed cumulant for the IVF in Section 3.1 only works for goal-reaching options, which although is quite general, it is not universal. For example, if the option’s task is to maximize the velocity of a robot and there is no specific target velocity, then we could not write down a 0/1 cumulant that faithfully describes that subtask. We will add this discussion to the camera-ready if our paper is accepted.

## New Experiments

- Reviewer uZYk asked how option horizon impacts initiation sets, Figure C in the PDF aims to answer this question.
- Reviewer zsHA was unclear about the contribution of TD to the IVF learning problem: Figure B in the PDF shows how TD is exploiting temporal structure for more sample efficient learning.
- Reviewer uZYk asked for visualizations of the initiation sets and the Initiation Value Function (IVF) in FourRooms; this is now included in Figure A of the PDF.

---

### Decision · Program_Chairs · 2023-09-21

**Decision:**

Accept (poster)

**Comment:**

This paper proposes a method for learning an initiation set for options framework. After identifying challenges in learning initiation sets, the paper introduces Initiation Value Function (IVF) which approximates the success probability of a given option in a given state and proposes to use it as a weight for training a binary option initiation classifier. The paper also proposes two types of exploration bonus when determining the initiation set to mitigate the pessimistic bias while learning options. The proposed method was evaluated in terms of the classifier's accuracy, option learning efficiency, and option discovery efficiency in various domains and tasks.

Most of the reviewers appreciated the structure of the paper (e.g., identifying/addressing challenges) and using a variety of domains for evaluation. However, several reviewers were confused by what exactly is the baseline or the proposed method (e.g., GVF), how exactly the evaluation was conducted (e.g., true initiation set, how to select horizon), why DSC is used for option discovery, and how the proposed method is combined with DSC, and so on. Although a few of them were clarified during the rebuttal period, two reviewers remained unconvinced even after the rebuttal, while one of the reviewers was willing to defend the paper for acceptance.

Since the reviewers have not come to an agreement, I read through the paper. Although I also found that the presentation of the empirical result is not great as pointed out by the reviewers, I believe that this paper has quite interesting ideas and findings that are worth to be presented. Thus, I recommend accepting the paper.

I strongly encourage the authors to improve the clarity of the empirical result section, by reflecting most of the reviewers' comments. Here is my additional suggestion:
- There are unnecessarily many combinations of the proposed method in each plot (Weighted, GVF, Optimistic in Fig 2, 3, and Weighted, GVF in Fig 5), and the explanation of each term is somewhat spread over the text. Instead, I would rather pick one variation and compare it against the baseline in the main figure and make a separate figure for ablation where many variations are compared.
- I do not see why it is necessary to show two variations of the exploration bonus instead one. They are not only distracting but also never compared with each other unless I missed it. I would suggest showing only one throughout the paper for clarity (and showing the other in the appendix if needed).